# Investigation of Cas9 antibodies in the human eye

Marcus A. Toral[1,2], Carsten T. Charlesworth[3], Benjamin Ng [1,4], Teja Chemudupati[1], Shota Homma[5], Hiromitsu Nakauchi [5,6,7], Alexander G. Bassuk [8], Matthew H. Porteus [3✉] & Vinit B. Mahajan [1,9✉]

Preexisting immunity against Cas9 proteins in humans represents a safety risk for CRISPR–Cas9 technologies. However, it is unclear to what extent preexisting Cas9 immunity is relevant to the eye as it is targeted for early in vivo CRISPR–Cas9 clinical trials. While the eye lacks T-cells, it contains antibodies, cytokines, and resident immune cells. Although precise mechanisms are unclear, intraocular inflammation remains a major cause of vision loss. Here, we used immunoglobulin isotyping and ELISA platforms to profile antibodies in serum and vitreous fluid biopsies from human adult subjects and Cas9-immunized mice. We observed high prevalence of preexisting Cas9-reactive antibodies in serum but not in the eye. However, we detected intraocular antibodies reactive to *S. pyogenes*-derived Cas9 after *S. pyogenes* intraocular infection. Our data suggest that serum antibody concentration may determine whether specific intraocular antibodies develop, but preexisting immunity to Cas9 may represent a lower risk in human eyes than systemically.

[1] Molecular Surgery Program, Department of Ophthalmology, Byers Eye Institute, Stanford University, Palo Alto, CA, USA. [2] Medical Scientist Training Program and Graduate Program in Molecular Medicine, University of Iowa, Iowa City, IA, USA. [3] Department of Pediatrics, Stanford University, Palo Alto, CA, USA. [4] Medical Sciences Division, University of Oxford, Oxford, UK. [5] Department of Genetics, Stanford University, Palo Alto, CA, USA. [6] Institute for Stem Cell Biology and Regenerative Medicine, Stanford University School of Medicine, Palo Alto, CA, USA. [7] Division of Stem Cell Therapy, Distinguished Professor Unit, Institute of Medical Science, University of Tokyo, Tokyo, Japan. [8] Departments of Pediatrics and Neurology and The Iowa Neuroscience Institute (INI), University of Iowa, Iowa City, IA, USA. [9] Veterans Affairs Palo Alto Health Care System, Palo Alto, CA, Palo Alto, CA, USA. ✉email: mporteus@stanford.edu; vinit.mahajan@stanford.edu

In CRISPR–Cas9 genome editing, the two most commonly-used Cas9 orthologs are derived from *Staphylococcus aureus* (SaCas9) and *Streptococcus pyogenes* (SpCas9)—two prevalent human commensal, and potentially pathogenic, microorganisms. Accordingly, patients eligible for CRISPR–Cas9 therapy may harbor antibodies reactive to Cas9 (α-Cas9) before treatment[1–3]. While there are discrepancies regarding the exact prevalence of these antibodies in humans, a majority of the general population may exhibit preexisting α-Cas9[3]. Furthermore, a high prevalence of preexisting cellular immunity against Cas9 (Cas9-reactive T-cells) has also been identified[2–4]. This preexisting immunity indicates that Cas9 protein expressed by edited cells may lead to immunological activation, inflammation, and potentially killing of edited cells—as has been demonstrated in vitro and in vivo[4–6]. This would represent a significant treatment barrier for CRISPR–Cas9 technologies. However, some tissues are not exposed to full systemic immunity and may not exhibit pre-existing Cas9 immunity. The eye, for example, demonstrates a degree of immune privilege. Nonetheless, inflammatory reactions to gene therapy in the eye have led to permanent vision loss for some patients[7–9].

The eye is currently at the forefront of genome editing, with a phase 1/2 clinical trial pioneering one of the first in vivo human trials for CRISPR–Cas9 therapy (identifier: NCT03872479)[10]. As the eye is considered an immune privileged organ, it is thought that ocular tissues may be less likely to mount inflammatory reactions to gene therapies. While ocular immune privilege remains incompletely understood, it has been shown that the eye exhibits physical barriers (e.g., blood-retina-barrier preventing free entry of systemic immune cells) as well as cell-bound and soluble immunosuppressive factors[11,12]. Yet, despite immune privilege, intraocular inflammation remains a major cause of vision loss—underscoring the limitations of our current understanding of the phenomenon. Indeed, while ocular immune responses are normally inhibited, inflammation may result if mechanisms become dysregulated, causing vision loss[13]. In the context of gene therapy, inflammatory reactions against adeno-associated viral (AAV) vectors used to deliver gene therapy components have been described. These reactions may be mediated, in part, by innate immune responses against the vector genome which attract CD8+ T-cells and drive inflammation[14]. However, a role for antibodies is increasingly being recognized[15]. Additionally, it is unclear whether reactions to Cas9 protein, derived from common ocular bacterial pathogens, would be mediated similarly. Indeed, the precise role of antibodies in ocular pathologies remains poorly understood. The ubiquity of antibodies in ocular tissues and reports of increased humoral levels of antibodies specific to gene therapy components after treatment warrants further research into the composition of intraocular antibodies and their potential for contributing to inflammation in the eye[7,15–18]. While systemic immune cells such as T-cells are normally absent from the eye, other immune components—including antibodies, cytokines, and complement molecules—maintain a constant presence[19,20]. All eye tissues except for the lens contain antibodies, and antibodies have been shown to be capable of mediating ocular pathology in multiple contexts[21–25]. It is possible that intraocular antibodies reactive to Cas9 protein could contribute to inflammation after retinal CRISPR–Cas9 treatment, or serve as useful biomarkers for predicting and monitoring inflammatory reactions.

Cas9 proteins, which function as endonucleases, are theoretically intracellular, unexposed to circulating antibodies. However, proteomic studies of the eye reveal that ocular tissues, including the retina, shed intracellular proteins into adjacent intraocular fluid compartments (e.g., the vitreous fluid; Fig. 1a)[20,26–28]. Additionally, sick cells that undergo editing may not survive, ultimately releasing cytosolic contents. Thus, even intracellularly-expressed Cas9 may represent a risk to the eye. If circulating antibodies within the eye exhibit preexisting reactivity to Cas9, this might lead to inflammation and vision loss. Specifically, Cas9 bound to intraocular antibodies may lead to subsequent activation of intraocular complement pathways and resident phagocytic cells, setting off a pro-inflammatory cascade which disrupts vision and could signal to systemic T-cells to infiltrate the eye[15,29]. Simply the potential for formation of antigen-antibody immune complexes represents a risk to the delicate intraocular environment[24,25,30,31]. While antibodies are a known component of the eye's immune system, little is known about their composition and no study has tested whether preexisting Cas9 antibodies circulate within ocular fluid compartments, such as vitreous fluid.

Vitreous fluid is a gelatinous extracellular matrix comprised primarily of water which sits between the lens and retina of the eye (Fig. 1a). This tissue makes up 80% of the human eye by volume and sequesters both intracellular and extracellular proteins shed from the retina and adjacent tissues[20,26–28]. Biopsies of vitreous fluid exhibit high enough intracellular and extracellular protein content to serve as proxy biopsies for the retina and guide treatment of human disease[20,26]. Furthermore, intraocular antibodies appear capable of limited transport between vitreous fluid and the retina[32]. Human vitreous fluid is thought to contain approximately 100-fold fewer antibodies than serum and it is unknown whether α-Cas9 is represented[21,33]. Given that current and future CRISPR–Cas9 therapies will rely on delivery and expression of bacterial Cas9 in the retina, an investigation into this question is particularly relevant.

Here, we test paired human vitreous fluid and serum biopsies (cohort 1: $n = 13$) from living adult subjects undergoing vitreoretinal surgery, and a validation cohort (cohort 2: $n = 36$) of unpaired human vitreous fluid biopsies, for preexisting α-Cas9 (Supplementary Table 1). Additionally, we immunize mice against Cas9 protein and test for α-Cas9 in the serum and vitreous fluid. In human vitreous fluid biopsies, we identify all subclasses of antibodies except for possibly IgM. We find that while preexisting α-Cas9 are highly prevalent in the serum, α-Cas9 are not detectable in the vitreous fluid except for in cases of severe disease or prior bacterial ocular infection (despite detection of control α-tetanus in nearly all vitreous fluid samples). In mice, we find that a subset of immunized mice developed vitreous fluid antibodies for each antigen tested. In both humans and mice, we observe that higher serum antibody levels trend with higher vitreous fluid levels of the same antibody, indicating that serum antibody levels may predict development of vitreous fluid antibodies. Our data indicate that preexisting immunity to Cas9 may represent a lower risk in human eyes than systemically, although further research is needed to conclusively rule out this risk.

## Results

**Characterization of human vitreous fluid antibodies.** First, we investigated the presence and distribution of immunoglobulin (Ig) subclasses in human vitreous fluid. Using an immunoglobulin isotyping panel, we measured levels of IgG₁, IgG₂, IgG₃, IgG₄, IgA, and IgM. To our knowledge, no study of vitreous fluid immunoglobulins to date has characterized IgG subclasses. It is unknown which antibody subclasses target Cas9, although predominantly IgG₁ and IgG₃ antibodies may be directed against *S. pyogenes* infection in human adults[34]. Our analysis detected vitreous antibodies at different concentrations based on subclass, detecting all subclasses except possibly IgM (Fig. 1b). This was expected, as IgM is the largest immunoglobulin and may be restricted based on size[21]. Notably, we observed higher vitreous

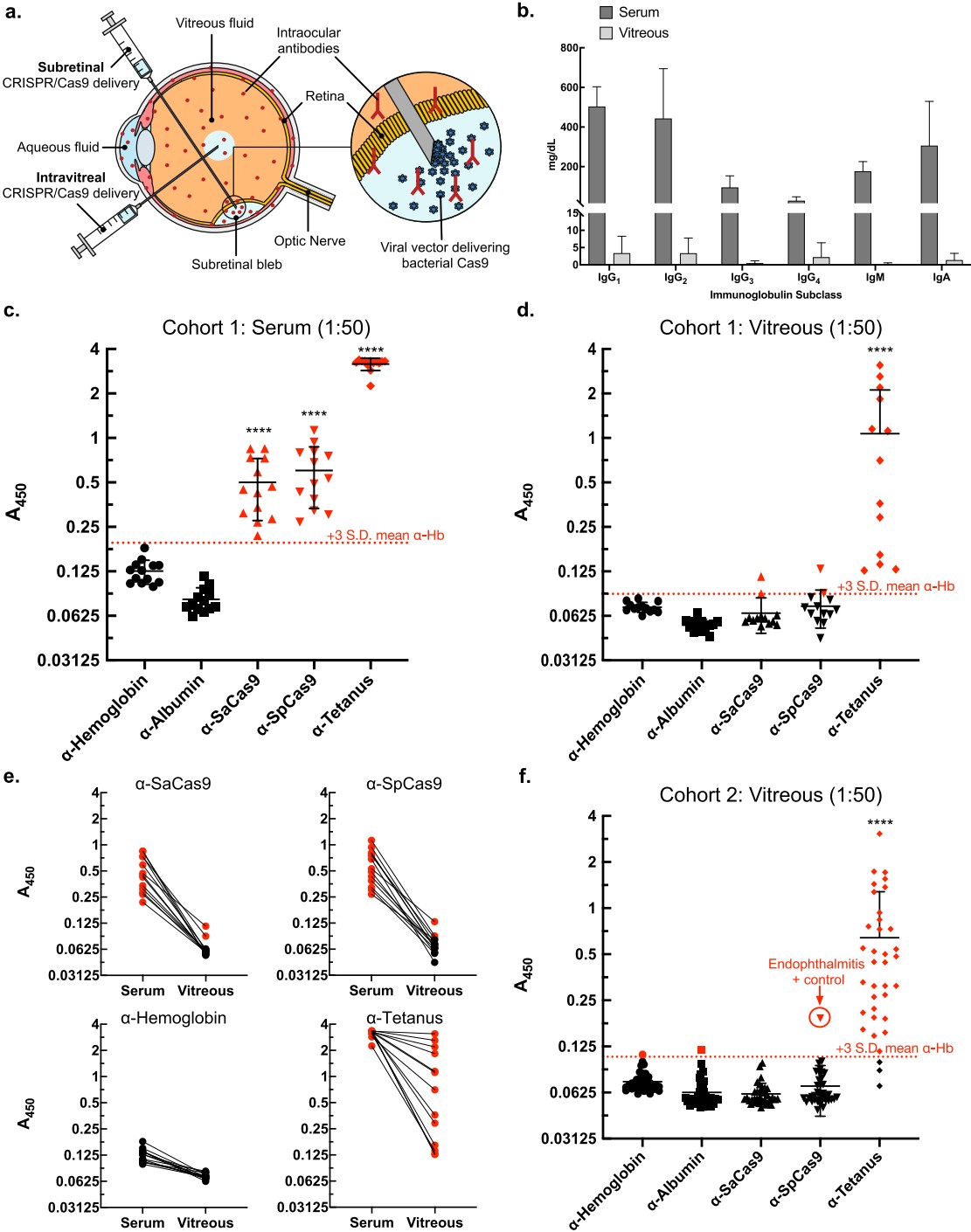

**Fig. 1 Preexisting Cas9 antibodies are prevalent in human serum but not vitreous fluid. a** Human retinal CRISPR–Cas9 gene editing is administered by subretinal or intravitreal injections. All eye tissues besides the lens contain antibodies (in red). **b** Estimated vitreous fluid immunoglobulin concentrations ($n = 26$) from isotyping detected all subclasses except possibly IgM, showing approximately 143-fold fewer total antibodies in aggregate compared to serum guideline ranges (from manufacturer). Bar heights represent mean average concentrations (mg/dL) and error bars represent range for each immunoglobulin subclass. **c** ELISA measurements from paired serum and D vitreous fluid samples (1:50 dilutions, $n = 13$) showed high α-Cas9 prevalence in the serum but not in vitreous fluid. Dotted red line indicates positivity cutoff value: mean α-hemoglobin A450 plus three standard deviations. Red symbols = positive, black symbols = negative. *S. pyogenes* or *S. aureus*-derived Cas9 denoted SpCas9 and SaCas9, respectively. **e** Analysis of paired samples showed that higher serum antibody levels generally corresponded to higher vitreous fluid antibody levels for each particular antibody. **f** ELISA measurements of vitreous fluid validation samples (1:50 dilutions, $n = 36$) confirmed very low α-Cas9 prevalence, though vitreous fluid from a patient with *S. pyogenes* intraocular infection (endophthalmitis) tested positive for SpCas9. For **c**, **d**, **f** error bars represent standard deviation. Anderson-Darling tests were performed to test for normal distribution of data. Statistical comparisons between normally distributed samples (α-SaCas9, α-SpCas9) used parametric unpaired two-tailed Student's t-tests, while similar comparisons between non-normally distributed samples (α-Tetanus) used non-parametric two-tailed Mann–Whitney U-tests. P-values < 0.05 were considered significant. Asterisks indicate p-value size: **** indicates p-value ≤ 0.0001. Source data are provided as a Source data file.

**Table 1 Comparison of mean ELISA A450 signals in serum and vitreous fluid.**

| Measurements | Charlesworth et al.[4] (Serum) | Present study (Paired serum) | Present study (Paired vitreous fluid) | Present study (Validation vitreous fluid) |
|---|---|---|---|---|
| Sample size | 125 | 13 | 13 | 36 |
| α-Tetanus positive samples | 99% | 100% (13/13) | 100% (13/13) | 92% (33/36) |
| α-Albumin positive samples | 0.8% | 0% (0/13) | 0% (0/13) | 3% (1/36) |
| α-Hemoglobin positive samples | Not shown | 0% (0/13) | 0% (0/13) | 3% (1/36) |
| α-SaCas9 positive samples | 78% | 100% (13/13) | 15% (2/13) | 0% (0/36) |
| α-SpCas9 positive samples | 58% | 100% (13/13) | 15% (2/13) | 3% (1/36) |
| α-Tetanus mean A450 | 2.057 | 3.162 | 1.070 | 0.640 |
| α-Hemoglobin mean A450 | Not shown | 0.125 | 0.072 | 0.074 |
| α-Albumin mean A450 | 0.340 | 0.081 | 0.054 | 0.063 |
| α-SaCas9 mean A450 | 1.340 | 0.500 | 0.065 | 0.061 |
| α-SpCas9 mean A450 | 1.092 | 0.602 | 0.073 | 0.069 |
| Threshold cutoff for positivity (Mean α-Albumin or α-Hemoglobin + 3 s.d.) | 0.879 (α-Albumin) | 0.195 (α-Hemoglobin) | 0.088 (α-Hemoglobin) | 0.107 (α-Hemoglobin) |

fluid $IgG_4$ levels than expected, finding that $IgG_4$ antibodies comprised ~20% of the total vitreous fluid antibodies, a much higher fraction than typically observed in the serum. In aggregate, we detected vitreous antibodies at a vitreous to serum ratio of approximately 1:143, with sample-to-sample differences. We observed no obvious correlations with patient diagnoses. For each individual antibody subclass, we detected the following approximate ratios of vitreous fluid to serum antibody concentrations, with vitreous antibody ranges and standard deviations reported in parentheses: $IgG_1 = 1:150$ (range: 19.05 mg/dL, s.d. 4.93), $IgG_2 = 1:133$ (range: 16.86 mg/dL, s.d. 4.43), $IgG_3 = 1:205$ (range: 2.69 mg/dL, s.d. 0.71), $IgG_4 = 1:13$ (range: 12.72 mg/dL, s.d. 4.19), $IgM = 1:926$ (range: 2.40 mg/dL, s.d. 0.45), $IgA = 1:230$ (range: 8.61 mg/dL, s.d. 1.98).

**Detection of α-Cas9 in human serum and vitreous fluid**. After detecting vitreous fluid antibodies, we next determined whether these antibodies included α-Cas9. Using an ELISA platform like that previously published[3], we tested paired vitreous fluid/serum samples for antibodies against human hemoglobin and albumin (negative controls), tetanus toxoid (positive control among the immunized human population), SaCas9, and SpCas9 (Fig. 1c, d). We applied a similar conservative threshold for positivity as previously published[3], defined as mean α-hemoglobin A450 value plus three standard deviations. In the serum, all 13 samples from our paired cohort were positive for α-tetanus. All 13 samples were also positive for α-SaCas9 and α-SpCas9. This 100% seropositivity was higher than previously reported[3], although this may be explained by differences in sample size or cohort demographics. In vitreous fluid from these same subjects, overall antibody levels were lower than in serum. Like serum, all 13 vitreous fluid samples were positive for α-tetanus. However, only two vitreous fluid samples tested positive for α-SaCas9 and two for α-SpCas9, showing antibody levels only near or below the lowest α-tetanus samples. Interestingly, looking only at the two positive samples with α-SaCas9/α-SpCas9 levels comparable to the lowest α-tetanus, we found that both samples came from the same patient (Supplementary Table 1, Case #8). This patient had been diagnosed with choroidal melanoma, a form of ocular cancer with the potential for major anatomical disruption to the eye and severe damage to the blood-retina-barrier with high potential for antibody leak from the serum into the vitreous fluid. Overall, we observed that higher serum antibody levels trended with higher vitreous fluid antibody levels, consistent with antibody diffusion from the serum into vitreous fluid even in cases without significant blood-retina-barrier disruption (Fig. 1e).

Next, we tested a validation cohort of human vitreous fluid biopsies ($n = 36$) representing diverse vitreoretinal diseases (Fig. 1f; Supplementary Table 1). We included one sample, as a positive control, from a subject diagnosed with *S. pyogenes* intraocular infection (endophthalmitis). Here, we detected 33 vitreous fluid samples positive for α-tetanus, no samples positive for α-SaCas9, and only one sample (the infectious endophthalmitis case) positive for α-SpCas9. These findings were summarized and compared to our previous serum study, overall demonstrating lower serum A450 measurements than previously published[3] except for α-tetanus (Table 1).

**α-Cas9 in serum and vitreous fluid of immunized mice**. To further address our findings that (1) the prevalence of α-Cas9 is reduced in vitreous fluid relative to serum, and (2) higher serum antibody levels predict higher vitreous fluid antibody levels, mice were immunized against ovalbumin (positive control), SpCas9, or SaCas9 via intramuscular injections of emulsified mixtures containing each purified protein antigen (Fig. 2a). We collected and analyzed mouse serum samples from immunized mice just prior to antigen injection (week 0) as well as 2, 4, and 6-weeks postinjection. Additionally, vitreous fluid was collected from mice at the final timepoint, 6-weeks post-injection, to pair with serum samples obtained from the same mice. There was robust, specific, and reliable immunization of mice against each protein by 6-weeks post-injection (Fig. 2b). Notably, SaCas9 immunization resulted in lower serum antibody levels compared with ovalbumin and SpCas9 in mice.

Finally, mouse vitreous fluid samples were collected and tested for development of ovalbumin or Cas9 antibodies following immunization. Interestingly, we observed that a subset of mice developed vitreous antibodies specific for ovalbumin (40% positive, Fig. 2c) and SpCas9 (44% positive, Fig. 2d), though this was reduced for SaCas9 (10% positive, Fig. 2e). Like the paired human samples, we observed a trend where higher serum antibody levels indicated higher vitreous fluid antibody levels across all immunization categories (Fig. 2f).

## Discussion

Taken together, these data indicate that in humans the prevalence of preexisting α-Cas9 in the eye is low despite high prevalence of preexisting serum α-Cas9. Should vitreous fluid antibodies represent an important inflammatory reactant or marker for overall Cas9-specific immune surveillance in the eye, this would bode well for current CRISPR–Cas9 clinical trials in the eye. However, identification of α-SpCas9 in vitreous fluid from a subject diagnosed with *S.*

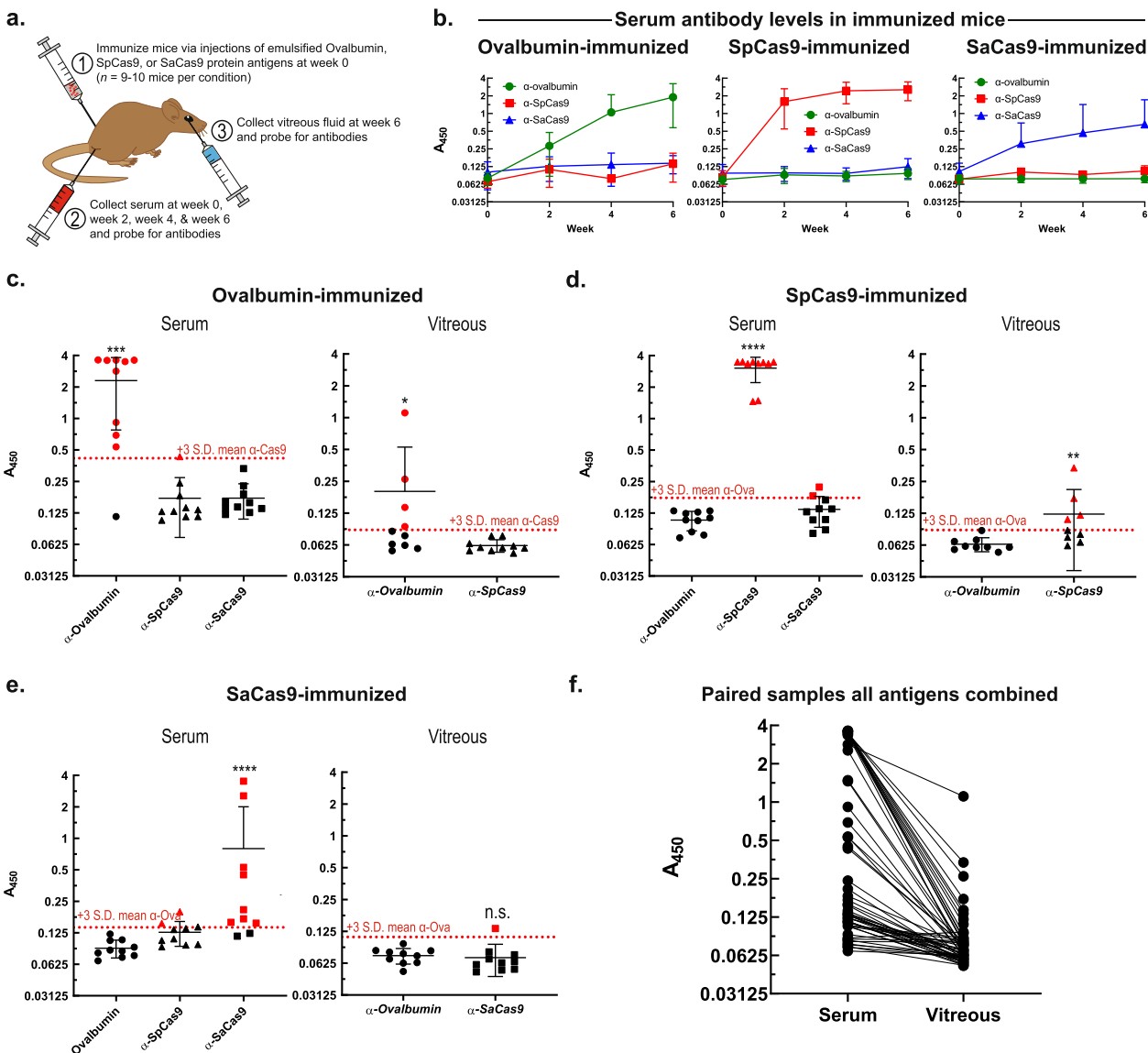

**Fig. 2 Mice immunized to Cas9 develop Cas9-specific antibodies in serum and in subset of vitreous fluid. a** Eight-week old C57BL/6J adult male mice were immunized against ovalbumin (positive control), SpCas9, or SaCas9. Serum and vitreous fluid samples were collected and probed for antibodies as described in Fig. 1. **b** At week 0 (prior to immunization), week 2, week 4, and week 6, serum was obtained from mice and levels of serum antibodies reactive to ovalbumin, SpCas9, or SaCas9 were measured by ELISA ($n = 10$ mice per condition). Green (circles), red (squares), and blue (triangles) datasets indicate mice immunized against ovalbumin, SpCas9, and SaCas9, respectively. Mean values shown, error bars indicate standard deviation. At week 6, vitreous fluid was also obtained from mouse eyes and paired to serum samples drawn at the same time from the same mice. Levels of serum and vitreous fluid antibodies reactive to ovalbumin, SpCas9, or SaCas9 were measured by ELISA and compared in (**c**) mice immunized against ovalbumin ($n = 10$ mice; serum [Ova vs. SpCas9] $p$-value = 0.0007, vitreous fluid $p$-value = 0.0297), **d** against SpCas9 ($n = 9$ mice; serum [SpCas9 vs. Ova] $p$-value ≤ 0.0001, vitreous fluid $p$-value = 0.0028), or **e** against SaCas9 ($n = 10$ mice; serum [SaCas9 vs. Ova] $p$-value ≤ 0.0001, vitreous fluid $p$-value = 0.2475). Data were checked for normality using Anderson-Darling tests and statistically significant differences were determined using two-tailed Mann–Whitney $U$-tests, with a significance level (α) of 0.05. Asterisks indicate $p$-value size: *, **, ***, and **** above data refer to $p$-values ≤0.05, ≤0.01, ≤0.001, and ≤0.0001, respectively. N.s. indicates no significant difference ($p$-value >0.05). Note that only two types of antibodies were tested for in vitreous fluid samples (one non-immunized negative control sample and one experimental sample) compared with three in serum samples (two non-immunized negative control samples and one experimental sample) due to limitations on the amount of vitreous fluid which could be obtained from mouse eyes. Dotted red line indicates positivity cutoff value: mean relative negative control A450 value (negative controls = non-immunized antigens) plus three standard deviations. Red symbols = positive, black symbols = negative. **f** Antibody levels of all paired serum and vitreous fluid samples were compared in each mouse across all immunization conditions, demonstrating a positive relationship between higher levels of antigen-specific serum antibodies and higher levels of the same antigen-specific vitreous fluid antibodies. Source data are provided as a Source data file.

*pyogenes* endophthalmitis shows that direct intraocular exposure to Cas9 may still increase α-Cas9 levels in vitreous fluid (Fig. 1f). Consequently, CRISPR–Cas9 surgeries in the eye may also have the potential to lead to the development of α-Cas9 in vitreous fluid. Our data show that testing vitreous fluid for antibodies directed against gene therapy components before, during, and after treatment is likely a safe and useful approach for monitoring patients in ongoing CRISPR–Cas9 clinical trials in the eye. Patients exhibiting high or

increasing levels of vitreous fluid antibodies might warrant exclusion from trials or closer monitoring for signs of developing intraocular inflammation. Given evidence of a link between particularly high increases in levels of serum antibodies reactive to gene therapy components (e.g., AAV vectors) and development of intraocular inflammation, it would be informative to see whether vitreous fluid antibodies show a similar trend[7,15–18]. Further studies are needed to investigate the immunological consequences of α-Cas9 in vitreous fluid (and, more broadly, in serum) for CRISPR–Cas9 gene editing. The use of mice may provide an important in vivo approach as new CRISPR–Cas9 technology continues to make its way into human medical therapy.

We observed that mice showed the weakest immunization against SaCas9, as demonstrated by antibody signals less than half of those seen for ovalbumin or SpCas9 by 6-weeks post-immunization (Fig. 2b). However, we observed that at least one mouse showed unusually high serum α-SaCas9 signal at the week 0 timepoint (prior to immunization) and generated a particularly rapid and robust antibody response (data not shown). This indicates that this mouse may have had preexisting immunity to SaCas9 and that preexisting Cas9 immunity led to an intensified immune reaction on repeat antigen exposure, although no clinical evidence of intraocular inflammation was observed. However, with a limited sample, further research is needed to evaluate this. With regards to the vitreous fluid, we observed that 40% and 44% of mice developed vitreous fluid α-ovalbumin and α-SpCas9, respectively, but only 10% developed vitreous fluid α-SaCas9 (Fig. 2c–e). Given our observation of lower serum α-SaCas9 levels post-immunization compared with α-SpCas9 and α-ovalbumin, we believe these differences are due to α-SaCas9 not reaching high-enough serum concentrations to diffuse into the eye more effectively. The exact mechanisms governing antibody entry and exit from the eye remain unclear, although our study provides new evidence that once levels of a particular antibody reach a high-enough serum concentration, they can overcome ocular barriers and enter vitreous fluid to reach detectable levels. This notion is further supported by evidence that higher serum antibody levels tend to predict higher levels of that same antibody in vitreous fluid in both humans and mice (Figs. 1e and Fig. 2f). Likewise, this would explain why α-tetanus (a product of deliberate, repeated immunization resulting in high serum antibody concentrations) but not α-Cas9 were highly prevalent in vitreous fluid (Fig. 1). Further research is needed to explore mechanisms of serum antibody entry into vitreous fluid.

In our human serum samples, it is notable that we detected such a high prevalence of preexisting α-Cas9, with 100% of donor samples found to be positive for α-Cas9 (Fig. 1c). This was higher than expected, particularly given that some studies have found preexisting serum α-Cas9 to be present in only 2.5–10% of large human cohorts[1,2]. The high prevalence we observed here may be due to the limited sample size of our paired cohort (cohort 1, $n = 13$) or demographic differences between cohorts. Regarding demographic differences, our paired cohort was older than others tested, with a mean age of about 66 years old (s.d. 13.9; Supplementary Table 1). It is unclear to what extent this difference may impact our results. On one hand, the older age of our cohort may be associated with a waning immunity. This may explain the lower raw ELISA A450 values we observed compared with previous work (Table 1)[3]. However, when these lower values were compared against our control groups, we found a higher overall rate of antibody positivity. On the other hand, the older age of our cohort may also be associated with increased risk or increased lifetime exposure to S. aureus or S. pyogenes, potentially leading to a higher prevalence of preexisting antibodies against these microbes.

Even with these considerations, however, the large discrepancies between studies regarding the prevalence of α-Cas9

may be due to relatively low levels of circulating α-Cas9 even in positive individuals, and differences in sensitivity between detection platforms used. In this regard, we believe that our platform is highly sensitive—a conclusion supported by evidence of our positive control α-tetanus signal reaching the detector maximum in serum samples (Fig. 1c). While this high sensitivity may indicate a potential for false-positives, the validity of antibody frequencies reported in our study is supported by the tractability of our system (e.g., including a positive control vitreous fluid sample from a patient with intraocular bacterial infection and seeing a corresponding increase in α-Cas9 levels, as shown in Fig. 1f), as well as patient-to-patient and clinical correlations in our data (e.g., higher serum α-Cas9 tending to predict higher vitreous α-Cas9; Fig. 1e, Supplementary Table 1). Notably, while there are discrepancies in the prevalence of α-Cas9 detected by different groups, multiple groups have reported high frequencies (>60% of patients) of Cas9 reactive T-cells in humans[2–4]. Indeed, this prevalence of cellular immunity is closer to that which we detect for α-Cas9, indicating regular immunologic exposure. Yet, despite the likely high sensitivity of our detection platform, we were still unable to detect high frequencies of α-Cas9 in vitreous fluid of the eye despite detecting high frequencies of control α-tetanus in the eye.

The relevance of α-Cas9 in serum compared to vitreous fluid may differ with regards to CRISPR–Cas9 gene therapy, reflecting the unique immunological environment and physiology of the eye. Unlike in the eye, systemic tissues are subject to regular surveillance by T-cells. Thus, preexisting T-cell populations may represent a greater risk to gene-edited systemic tissues than antibodies at any prevalence or concentration[4–6]. However, systemically, a risk still exists for the formation of immune complexes consisting of Cas9 antigen bound to preexisting circulating antibodies, a well-described phenomenon in a variety of human diseases[35]. This risk would likely depend on the amount of Cas9 being expressed by edited cells, the duration of expression needed to achieve therapeutic efficacy, and the potential for extracellular Cas9 leak from edited tissue.

If immune reaction to SaCas9/SpCas9 proves to be an ongoing hurdle, there are multiple routes currently under development to attempt to circumvent this. One option is to use Cas9 protein derived from microbes that do not colonize humans, such as the thermophilic bacterium Bacillus hisashi[36] or, similarly, Geobacillus stearothermophilus[37]. Another option may be to engineer forms of Cas9 that are designed to be less immunogenic through silencing of immunodominant epitopes[2], or through inclusion of oligonucleotides which directly antagonize relevant immune receptors[14]. Separately, an additional route to overcome this issue might involve inducing immune tolerance, perhaps through methods of expanding the Cas9-specific regulatory T-cell population[4,38]. In the eye, the most straightforward approach is to include strong ocular immunosuppressive therapies and careful patient monitoring[16].

With these points all taken into consideration, there are limits to the present study. First, as vitreous fluid was obtained during surgery, human subjects in our cohorts suffered from varying degrees of eye disease (vitreoretinopathies). Yet, even with these pathologies, we observed low intraocular α-Cas9. This bodes well for retinal CRISPR–Cas9 therapy, as those eligible also suffer from pre-treatment vitreoretinopathies. Indeed, two subjects in our study (cases #11 and #12; Supplementary Table 1) were diagnosed with cone-rod dystrophies, conditions similar to Leber congenital amaurosis (the focus of the ongoing aforementioned clinical trial which utilizes SaCas9[10]), and tested negative for vitreous fluid α-SaCas9 despite demonstrating serum α-SaCas9. While surgical delivery of gene therapies in the eye (particularly subretinal injection compared with intravitreal injections) may

involve some degree of traumatic disruption to ocular anatomy, our data indicate that only severe blood-retina-barrier disruption may lead to increases in vitreous fluid α-Cas9, as seen in the sample from our subject who exhibited detectable vitreous fluid α-Cas9 in the setting of ocular cancer (Supplementary Table 1, Case #8). A second limitation of our study involves the difficulty in determining the threshold for calling a vitreous fluid sample positive. For consistency, we chose a similar stringent cutoff as previously reported in the serum[3]. However, despite indicating statistical rarity, these cutoffs may be biologically arbitrary. Furthermore, while ELISAs are sensitive, it is possible that vitreous fluid samples contained α-Cas9 below detection capabilities. It is unclear at what level vitreous fluid antibodies become clinically impactful, though even low levels might exert functional consequences[39].

Prior reports of adverse immune reactions to gene therapies in the eye warrant careful safety considerations. We confirmed that antibodies against bacterially-derived Cas9 are prevalent in human serum. However, we found that these antibodies were typically not detectable in the human eye without prior ocular exposure to Cas9 antigen (e.g., bacterial eye infection) or severe ocular pathology, and even then, they were observed at relatively low levels. Nonetheless, data from mice and humans indicate that high levels of serum α-Cas9 may predict development of detectable vitreous fluid levels of α-Cas9. Our data indicate the same is likely true for any systemically circulating antibody, though antibody subclass may also play a role. Overall, while these findings bode well for current and future intraocular CRISPR–Cas9 applications, further research is needed to determine whether intraocular Cas9 exposure from CRISPR–Cas9 therapy increases intraocular α-Cas9, and to what extent α-Cas9 can impede treatment in the eye or predict inflammation.

## Methods

**Study approval**. The study obtained Stanford University Institutional Review Board approval and adhered to the tenets set forth in the Declaration of Helsinki. Written informed consent was obtained from all participants, explaining the nature and purpose of the study. Tissue samples were collected between September 2019 and March 2020. All mouse experiments described were conducted in accordance with approved Stanford University IACUC protocols and the ARVO Statement for the Use of Animals in Ophthalmic and Vision Research.

**Human vitreous and serum biopsy collection**. Vitreous fluid biopsies were obtained from patients undergoing vitreoretinal surgery with vitrectomy for a variety of surgical indications (Supplementary Table 1). Vitreous fluid was obtained by pars plana vitrectomy, performed using a single-step transconjunctival 25-gauge trocar cannular system (Alcon Laboratories Inc, Fort Worth, TX), and an undiluted 1-cc sample of vitreous fluid was manually aspirated into a 3-cc syringe[40,41]. Vitreous samples were immediately centrifuged in the operating room at $15,000 \times g$ for 5 min at room temperature to remove impurities and then stored at $-80\,°C$[42]. Serum samples were collected by whole blood draws into EDTA tubes to prevent coagulation and kept at 4 °C for no more than 24 h. To separate the serum, whole blood samples were transferred to polypropylene tubes and let sit at room temperature for 30 min to allow clot formation. Clots were removed by centrifugation at $1500 \times g$ for 10 min at 4 °C. The resulting supernatant (serum) was collected and stored at $-80\,°C$ until use. For paired serum/vitreous biopsies (cohort 1), vitreous fluid and serum were collected within one hour of one another.

**Human immunoglobulin isotyping panel**. Immunoglobulin isotyping was performed through the Stanford Human Immune Monitoring Center (http://iti.stanford.edu/himc.html) using a Milliplex Human Isotyping Magnetic Bead Panel Multiplex Assay (Millipore Sigma) to test for concentrations of $IgG_1$, $IgG_2$, $IgG_3$, $IgG_4$, IgA, and IgM in a subset of vitreous biopsies from cohort 2 ($n = 26$). This subset did not include all of cohort 2 due to volume limitations for some vitreous fluid samples and prioritization of samples for ELISA. A 1:40 dilution of vitreous fluid was experimentally determined to be most appropriate. Five parameter logistic regression in MasterPlex (Hitachi Software) was used to predict concentrations of antibodies in vitreous samples from raw mean fluorescence intensity (MFI) values[43]. Specific samples included in the isotyping cohort, as well as raw MFI and predicted antibody concentrations, are available in the Source Data file. Sample numbers may be linked to patient cases in Supplementary Table 1.

**ELISA**. ELISA platform and protocol was adopted from Charlesworth, et al.[3]. Greater than 95% pure SpCas9 and SaCas9 proteins were provided by Integrated DNA Technologies (IDT). Tetanus toxoid was purchased from Astarte Biologics, and human hemoglobin and human albumin were purchased from Sigma-Aldrich. Protein antigens were coated onto a 96-well Maxisorp plate (Thermo Fisher Scientific) overnight at 4 °C in 1× bicarbonate buffer (Sigma-Aldrich). Plates were then blocked with 5% skim milk (BioRad) for 2 h at room temperature. Vitreous and serum samples were diluted 1:50 in 5% skim milk; plates were incubated overnight at 4 °C. Plates were then washed $3 \times 5$ min[3]. HRP-conjugated goat anti-human Fc antibody (Bethyl Laboratories) was then applied at a dilution of 1:100,000 in 5% skim milk and incubated for 1 h at room temperature. 3,3',5,5'-Tetramethylbenzidine substrate solution (Thermo Fisher Scientific) was then added and allowed to develop for 20 min before 1 N sulfuric acid (Thermo Fisher Scientific) was added to stop the reaction. The absorbance at 450 nm (A450) was then analyzed using a SpectraMax M3 microplate reader (Molecular Devices). ELISAs were performed in experimental triplicates and the mean average measurements from those three experiments are shown. Raw ELISA A450 values are available in the Source data excel file.

**Mouse Immunization and ELISA**. 8-week-old adult male C57BL/6J mice were obtained from Jackson Labs. An emulsion of protein antigen (Ovalbumin [Sigma-Aldrich], SaCas9 [IDT], or SpCas9 [IDT]) and Freund's Adjuvant was prepared and 25 µg of antigen in Freund's Adjuvant was then injected intramuscularly into each thigh (50 µg total) ($n = 10$ per group). At the time of injection, as well as 2, 4, and 6-week post-injection, serum was collected. Additionally, at the final timepoint 6-weeks post-injection, vitreous fluid was collected via anterior segment dissection followed by lens evisceration, retina and vitreous evisceration, and filtered centrifugation of eviscerated retina and vitreous at $14,000 \times g$ for 12 min[44]. The resulting eluant in the lower chamber, representing the vitreous fluid, was then aspirated and collected. Subsequently, serum and vitreous fluid samples were analyzed by ELISA as described above, instead using HRP-conjugated goat anti-mouse Fc antibody (GE Healthcare) applied at dilution of 1:10,000, probing for antibodies against ovalbumin (positive control), SaCas9, and/or SpCas9. Statistical analyses were performed as described below, using antibodies in non-immunized mice as negative controls for comparisons (e.g., In mice immunized to SpCas9 antigen but not Ovalbumin antigen, α-SpCas9 levels were compared to α-Ovalbumin levels as a negative control, as mice were Ovalbumin-naïve). Mouse raw ELISA A450 values from both serum and vitreous are provided in the Source Data excel file.

Mice were housed in Stanford University institutional animal care and use committee (IACUC)-approved facilities, providing 12 h/12 h light/dark cycles, temperatures of 18–23 degrees Celsius, and 40–60% humidity. Food and water were made available at all times and handling, noise and vibrations, and odors were kept to a minimum. A veterinarian was on call 24 h per day and animals were observed daily by vivarium staff to ensure adequate health and welfare. All mouse experiments described were conducted in accordance with approved Stanford University IACUC protocols and the ARVO Statement for the Use of Animals in Ophthalmic and Vision Research.

**Statistical analyses**. Five-parameter logistic regression was performed in MasterPlex (Hitachi Software) to predict antibody concentrations from raw MFI immunoglobulin isotyping data[43]. To check for normal distribution of ELISA data, we performed Anderson-Darling tests. Statistical comparisons between normally distributed samples used parametric unpaired two-tailed Student's $t$-tests, while similar comparisons between non-normally distributed samples used non-parametric two-tailed Mann–Whitney $U$-tests. Analyses were performed to test for statistically significant differences ($p < 0.05$) between α-Hemoglobin (negative control) and α-Tetanus, α-SaCas9, or α-SpCas9. All statistical analyses for ELISA data were performed in PRISM 9.3.1 (GraphPad Software).

**Reporting summary**. Further information on research design is available in the Nature Research Reporting Summary linked to this article.

## Data availability

All data analyzed or generated during this study are available within the article in the Source Data or Supplementary Information files. Source data are provided with this paper.

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

## Acknowledgements

V.B.M. and A.G.B. are supported by NIH grants [R01EY031952, R01EY030151, R01NS98950, R01EY025225, R01EY024698, P30EY026877], and Research to Prevent Blindness (RPB) New York, NY. M.A.T. is supported by NIH grant [T32GM007337]. V.B.M. and M.H.P. are supported through the Stanford University Center for Definitive and Curative Medicine (CDCM). M.H.P. is supported by the Amon Carter Foundation and the Laurie Kraus Lacob Faculty Scholar Award in Pediatric Translational Research for this work. C.T.C. is supported by the National Science Foundation Graduate Research Fellowship under Grant No. (DGE-1656518). The funding organizations had no role in design and conduct of the study; collection, management, analysis, and interpretation of the data; preparation, review, or approval of the manuscript; and decision to submit the manuscript for publication.

## Author contributions

Vinit B. Mahajan had full access to all the data in the study and takes responsibility for the integrity of the data and the accuracy of the data analysis. Study concept and design: M.A.T. and V.B.M. Acquisition of data: M.A.T., V.B.M., C.T.C., T.C., and S.H. Analysis and interpretation of data: M.A.T., C.T.C., M.H.P., and V.B.M. Drafting of the manuscript: M.A.T., B.N., and V.B.M. Critical revision of the manuscript for important intellectual content: H.N., A.G.B., M.H.P., and V.B.M. Statistical analysis: M.A.T. and V.B.M. Obtained funding: V.B.M. Administrative, technical, and material support: M.H.P. and V.B.M. Study supervision: V.B.M. Thanks to Jing Yang for collection of mouse vitreous fluid samples.

## Competing interests

V.B.M is a co-founder and holds equity Eurdora and serves as an advisor for which he receives compensation from Replay. M.H.P. is a co-founder and Board member of Graphite Bio and holds equity in CRISPR Tx. He has equity and serves on the Scientific Advisory Board of Allogene Tx and Ziopharma Tx. He serves as an advisor for which he receives compensation from Versant Ventures. These companies had no input into this work. H.N. serves on the Scientific Advisory Board and holds equity in Century Therapeutics, QihanBiotech, and Megakaryon Corp, but these companies had no input into this work. The remaining authors declare no competing interests.
