## [Peer Review File · Nature Communications]

Reviewers' Comments:

Reviewer #1:

Remarks to the Author:

The group investigated to what extent Cas9 immunity is relevant in the eye, which will be the first organ treated by in vivo CRISPR-Cas9 clinical trials. The group finds antibodies in serum directed against Cas9, but not in the eye. However, when the eye was infected with *S. aureus*, the authors found intraocular antibodies against *S. aureus*-derived Cas9.

This is a very timely article (first patients dosed with AAV-SaCas9) and the data is novel, presented in a very clear manner, especially in the result section.

Critique 1: conceptually, the role of a-Cas9 antibodies may aggravate inflammatory responses and prior literature in ocular gene therapy has shown dose-dependent inflammation triggered by AAV alone - most probably due to innate immune responses to the vector genome which attract CD8+ T cells. Therefore, the detection of a-Cas9 antibodies in the eye only poses a theoretical risk, while preexisting Cas9-specific T cells would be more detrimental.

Critique 2: The discussion seems circular and repetitive and there is little discussion of prior data on Cas9-immunity by other groups (only references to own work), in particular no comment on discrepancy between a-Cas9 antibody prevalence between original Cas9-immunity paper by this group and prior work by Simhardi et al (PMID: 30073181).

I rather recommend the work for a brief report (more suitable as „brief communication“ instead of a full article).

Specific comments:

Abstract:

- line 39/40: authors infer a logical connection between antibodies and ocular inflammation after gene therapy, which is not established as a mechanism in the field to my knowledge.
- line 45/46: how does de novo intraocular Cas9-immunity arise? Not discussed in the paper: immune privilege = little amounts of adaptive immune cells around, please delete this comment, because it is overestimating the role of humoral anti-Cas9 immunity and is not in line with current understanding how AAV elicits inflammation (which is the activation of innate immune response by vector genomes, which then attracts T cells specific for vector).

Introduction:

- line 52/53: authors only refer to own work, not to the results from Simhadri paper, where significantly less a-Cas9 antibodies were reported (5%?)
- line 53/54: cellular immunity was also reported in Wagner et al 2019 (PMID: 30374197) and Ferdosi et al 2019 (PMID: 31015529)
- line 54-65: Wagner et al (PMID: 30374197) showed killing of SpCas9-overexpressing cells by Cas9-specific T cells in vitro, please cite instead of review article (??).
- line 59-61: AAV elicits dose dependent inflammation in ocular gene therapy trials indicating a particular role of innate immune pathways (e.g. cite Chan et al 2021 PMID: 33568518, see Suppl. Fig. 18+19 for results from systemic literature review)
- line 63/64: please cite original work by Maeder et al (PMID: 30664785), which is the basis of this first clinical trial
- line 76/77: true, but is this really mechanistically established?
- line 79: true + good comment, but toxicity in ocular gene therapy is dose-dependent and relevance of innate immunity is already established (innate immune activation → CD8 T cells lead to elimination of retinal cells)
- line 89/90: what about rate of anti-AAV antibodies or other anti-vector antibodies? Has there been prior literature that directly links preexisting antibodies to ocular inflammation? → ref 15 cited in the discussion just looks at the prevalence in corpses.

Discussion:

- would it be possible to use the method described in this paper to evaluate patients undergoing ocular gene therapy? If yes, this could potentially establish the link between anti-Cas9/AAV

antibodies and excessive inflammation in the eye after gene therapy.

- the authors must discuss why they detect serum a-SpCas9 antibodies at such a high rate, while Simhadri et al detected only 2.5% and Ferdosi et al 8.8% in much larger cohorts? Simhadri reported higher rates for a-SaCas9 ab (10%)

- what is the expected impact of the surgery required for injection of ocular gene therapy? Could transfer of aCas9 antibodies appear later due to lost barrier function?

Reviewer #2:

Remarks to the Author:

Nat Communications (NCOMMS-21-08338)

Cas9 Antibodies in the Eye

Toral et al. have studied the preexisting immunity against Cas9 proteins in human serum and vitreous fluid samples.

Congratulations to the authors for providing a highly relevant and intriguing study investigating the presence of Cas9 antibodies in human eyes. The study is an extension of a previous study identifying preexisting adaptive immunity to Cas9 proteins in humans. By means of immunoglobulin isotyping and ELISA the authors elegantly show high prevalence of Cas9-reactive antibodies in serum samples but not in vitreous fluid samples. The findings are sound and the manuscript is well-written. Further to this, the presented results are interesting for the society and may bode for novel treatment application based on CRISPR-Cas9. However, as pointed out further down below it contains a number of issues which need to be addressed before acceptance can be warranted.

Major concerns

1. Age of donors is significantly higher compared to the median age of donors tested in Ref 1. In the latter the median age of donors tested was 43. This difference should be taken into consideration in the conclusions.
2. The authors need to further substantiate the notion that high enough serum concentrations may become detectable in the eye. This may easily be validated in mice experiencing sepsis due to e.g., *S. aureus*.
3. In would strengthen the paper to include analysis of immune reactions (paired vitreous-serum biopsies) in e.g., mice (with or without preexisting Cas9 antibodies) following viral vector-based ocular delivery of clinically relevant levels of Cas9. The delivery route (subretinal or intravitreal) may also have impact on the immune response.
4. As stated, up to 58-78 % of the general population may exhibit preexisting Cas9 ab. Even though the focus of the present study is the eye, please include a brief discussion on the high abundancy of preexisting Cas9 antibodies in relation to the relevance of Cas9-based gene therapy in tissues that are not immune privileged.

Minor concerns

1. The title is misleading since the main finding is that Cas9 antibodies apparently are not present in the eye. Please rephrase and include "human". Alternatively, include "Human" and "?" in the submitted version: Cas9 Antibodies in the Human Eye?
2. Is Reference NCT02168686 correct? This is a gene therapy trial to treat A1AT deficiency. Maybe I missed it, but it is not obvious from the web page how the authors conclude from this reference that immune reactions to gene therapy in the eye have halted trials. Please verify.
3. Page 5, Results. Please indicate which cohort the sample originate from. Moreover, the "number of vitreous biopsies" is used inconsistently: In the Figure 1 legend it listed as 26, in the Supplementary Methods, it is 28. Please verify.
4. Line 99, "approximately 150-fold fewer". From Figure 1B up to 200-fold fewer total antibodies can be observed. Range should be indicated.
5. For clarity use vitreous fluid in the manuscript. Not intraocular fluid.
6. Regarding Suppl Table 1: Please stratify the cohort information, e.g., add some descriptive features as gender ratio and median age.

7. Will the modified CRISPR/Cas tools, which have emerged or are on its way, attract attention from the immune system similar to SaCas9 and SpCas9?

Response to Reviewer #1 (Remarks to the Author):

General critiques:

1. Critique 1: Conceptionally, the role of a-Cas9 antibodies may aggravate inflammatory responses and prior literature in ocular gene therapy has shown dose-dependent inflammation triggered by AAV alone - most probably due to innate immune responses to the vector genome which attract CD8+ T cells. Therefore, the detection of a-Cas9 antibodies in the eye only poses a theoretical risk, while preexisting Cas9-specific T cells would be more detrimental.

We thank the reviewer for this insightful comment and agree that detection of preexisting Cas9-specific T-cells in the eye would represent a significant risk to edited retinal cells, and that T-cells may be primary effector cells in some retinal immunopathologies including AAV-triggered inflammation. However, T-cells are normally absent from the eye, in particular from the vitreous fluid and subretinal space, and their specific role in various retinal inflammation conditions is controversial. On the other hand, antibodies are normally circulating in the vitreous fluid and sample the retina space, and preexisting antibodies to antigens are a well-established concern. The contribution and association of antibodies to retinal pathology has been described in multiple contexts, including AAV-related inflammation. While exact mechanisms of antibody function in the eye are unclear, the potential for formation of immune complexes consisting of Cas9 antigen bound to preexisting Cas9 antibodies represents at least one inflammatory mechanism which threatens the delicate intraocular environment independent of preexisting T-cells.

With regards to harmful consequences of antibodies in the eye, studies of autoimmune retinopathy have shown that exposing retinal cells to autoantibodies *in vitro* can induce their apoptosis, and a similar phenomenon has been observed *in vivo* following intravitreal injection of autoantibodies into rat eyes (PMID: 17235687, PMID: 12848960). While anti-Cas9 are not necessarily akin to autoantibodies, this finding demonstrates the immunopathological risk of intraocular antibodies independent of T-cells. Furthermore, formation of deleterious extracellular antibody-antigen immune complexes has been demonstrated in the context of eye disease. This mechanism is more closely relevant to potential inflammatory consequences of anti-Cas9, whereas anti-Cas9 might form immune complexes with extracellular Cas9 in the vitreous, shed from edited cells. Indeed, the presence of antibody-antigen immune complexes is strongly associated with development of uveitis in humans (PMID: 27428230), and immune complexes are thought to play a significant and direct role in the pathology of lens-induced endophthalmitis, for example (PMID: 1566234). Furthermore, in the context of age-related macular degeneration, immune complex formation may play a significant role in the development of drusen (PMID: 10865992). Thus, recognition of ocular Cas9 antigen by preexisting antibodies risks formation of harmful immune complexes. These immune complexes have the potential to bind Fc γ receptors expressed on the surface of microglia and other retinal cells, driving inflammation. Accordingly, intravitreal injection of antigen in immunized mice has been shown to generate antibody-antigen immune complexes throughout the retina, including large deposits in the subretinal space, leading to a potent inflammatory response involving activation of microglia, macrophages, and expression of pro-inflammatory cytokines (PMID: 24334446). These antibody-dependent inflammatory mechanisms do not involve T-cells.

While T-cells are thought to be drivers of intraocular inflammation, they are normally absent from the eye—particularly the posterior compartment where the retina is located. On the other hand, antibodies, cytokines, complement molecules, and resident phagocytic cells have a permanent presence in the eye. Thus, it is thought that the main function of intraocular antibodies is to clear infections and limit the spread of pathogens while innate and adaptive cell-mediated immunity remains suppressed (PMID: 12852492). Yet, while antibodies appear capable of inducing intraocular inflammation independent of T-cells, antibody-mediated inflammation is likely also synergistic with T-cells, facilitating T-cell infiltration and potentiating their activities. Studies in mouse models of autoimmune retinopathy have shown that even when the blood-retina-barrier is experimentally broken to facilitate T-cell entry, removal of antibodies results in slower disease progression and a milder form of the disease (PMID: 30635390). Furthermore, while AAV-mediated inflammation can be driven by T-cells, studies have reported a link between systemic AAV antibody levels and episodes of intraocular inflammation, including vitritis and uveitis (PMID: 25938638, PMID: 30730541, PMID: 28526489, PMID: 28647203). Accordingly, even if anti-Cas9 itself does not directly drive disease, its presence may still serve as a useful and novel biomarker for predicting the potential for immune response. This is also supported by the strong correlation between the presence of anti-retinal antibodies and the development of autoimmune retinopathy, even though the disease is mediated by T-cells (PMID: 24315290)

Ocular immune responses are complex and remain incompletely understood, but they are likely mediated by multiple inflammatory mechanisms. Evidence indicates that T-cells are key drivers of ocular inflammation, but the role of antibodies and immune complexes in the eye is particularly poorly understood. The ubiquity of antibodies within ocular tissues and evidence of a direct pathological role in some contexts indicates that they are an important component to consider and likely also serve as a sensitive biomarker for gauging overall immunosurveillance.

Based on the reviewer's important comment, we have updated the text to better reflect this perspective, the limitations of our study, and recognition of the potential role of T-cells.

2. Critique 2: The discussion seems circular and repetitive and there is little discussion of prior data on Cas9-immunity by other groups (only references to own work), in particular no comment on discrepancy between a-Cas9 antibody prevalence between original Cas9-immunity paper by this group and prior work by Simhardi et al (PMID: 30073181).

We agree with the reviewer's astute comment that this discrepancy is critical to discuss, and we thank them for this suggestion. As such, we have updated the text of the Discussion to comment on these discrepancies in the context of the field and have included references to other important studies in the field, including the work by Simhardi *et al.* and have updated the discussion to remove repetitive text, as described below.

The reviewer raises an important point regarding discrepancies between studies investigating the prevalence of anti-Cas9 in human donors. While data from our detection platform show much higher anti-Cas9 prevalence than the work by Simhardi *et al.*, and we have added this reference, we believe these differences may be due to higher sensitivity of our platform. Furthermore, our results show high correlation with clinical data and remain closely aligned with reported frequencies of cellular immunity in donor samples, supporting our conclusions.

Using our platform, previous work detected antibodies against *S. pyogenes* Cas9 (SpCas9) and *S. aureus* (SaCas9) in 58% and 78% of human donor serum samples, respectively (PMID: 30692695). In the present study, we again identified high prevalence of these antibodies, with 100% of our 13 paired serum samples positive for both anti-SpCas9 and anti-SaCas9. These results differ from those in the mentioned study by Simhardi *et al.* (PMID: 30073181) as well as a study by Ferdosi *et al.* (PMID: 31015529), where only 2.5% (10% for SpCas9) and at least 5% of donor serum samples were called positive for anti-Cas9, respectively. However, both Ferdosi *et al.* and Wagner *et al.* detected T-cells reactive to Cas9 at frequencies of 60% or higher, indicating that the human immune system is exposed to and develops a response to Cas9 regularly (PMID: 31015529, PMID: 30374197). With regards to circulating anti-Cas9 levels, we believe that large differences in prevalence reported between studies are due to relatively low levels of circulating anti-Cas9 (even in positive individuals) and differences in sensitivity between detection platforms. We added this explanation to the discussion text.

In this regard, we believe that our platform is more sensitive—a conclusion supported by evidence of our positive control anti-tetanus signal maxing out the detector in our serum samples (Fig 1C). While this high sensitivity may indicate the potential for false-positives, the validity of antibody frequencies reported in our study is supported by the tractability of our system (e.g., including a positive control vitreous sample from a patient with intraocular bacterial infection and seeing a corresponding increase in anti-Cas9 levels, as shown in Fig 1F), as well as patient-to-patient and clinical correlations in our data (e.g., higher serum anti-Cas9 tending to predict higher vitreous anti-Cas9 [Fig 1E]). With regards to the latter point, in Fig 1D, both positive anti-Cas9 vitreous samples came from the same patient with ocular cancer (Supplemental Table 1, Case #8)—a condition showing major anatomical disruption to the eye and severe damage to the blood-vitreous barrier. As expected, this patient showed higher anti-Cas9 antibody levels, likely representing significant leak from the serum into the vitreous, a finding which was observed due to the sensitivity of our platform.

Furthermore, our reported frequencies of anti-Cas9 in human donor serum are more closely aligned with reported frequencies of Cas9-reactive T-cells (cellular immunity) in human donor serum. This consistency with reported frequencies of cellular immunity strengthens the validity of our platform. Specifically, Ferdosi *et al.* found Cas9-reactive T-cells in the majority of their healthy cohort (PMID: 31015529) and Wagner *et al.* found 96% of their donor samples to be positive for Cas9-reactive T-cells (PMID: 30374197). Thus, while mechanisms of cellular and humoral immunity differ, it is not unrealistic to conclude that antibody frequencies may occur at similar levels. Of course, further research by independent groups is needed to fully resolve these discrepancies in antibody frequencies.

Finally (and importantly), despite the likely high sensitivity of our detection platform, we were unable to detect high frequencies of anti-Cas9 in the vitreous of the eye despite detecting high frequencies of control anti-Tetanus antibodies in the eye. This supports the core overall conclusion of the present work that preexisting immunity to Cas9 in the serum does not occur at the same frequency in the eye.

3. Critique 3: I rather recommend the work for a brief report (more suitable as “brief communication“ instead of a full article).

Per both reviewer’s suggestions, we have significantly expanded the text, as well as added an additional mouse dataset and figure along with additional supplemental data to the manuscript (Figure 2, Supplemental Data 3). We will work with the journal editors to determine the best format.

Critiques pertaining to the Abstract:

4. Critique 4: Line 39/40: authors infer a logical connection between antibodies and ocular inflammation after gene therapy, which is not established as a mechanism in the field to my knowledge.

This is a fair point, as mechanisms of antibodies in the eye are poorly understood, and we have adjusted the words to acknowledge this. However, multiple studies have reported a correlation between levels of neutralizing antibody levels to AAV gene therapies and development of intraocular inflammation (PMID: 25938638, PMID: 30730541, PMID: 28526489, PMID: 28647203). Furthermore, we believe there is a logical connection between antibodies and ocular inflammation to Cas9 protein, a bacterial protein that does correlate with a case of intraocular infection, rather than inflammation specific to gene therapy more broadly. The Cas9 protein is derived from a known immunogenic ocular pathogen. Given the role of antibodies in other ocular pathologies (see response to Critique #1, above), we believe that our study adds value and a new perspective to the field through interrogation of these antibodies.

5. Critique 5: Line 45/46: how does de novo intraocular Cas9-immunity arise? Not discussed in the paper: immune privilege = little amounts of adaptive immune cells around, please delete this comment, because it is overestimating the role of humoral anti-Cas9 immunity and is not in line with current understanding how AAV elicits inflammation (which is the activation of innate immune response by vector genomes, which then attracts T cells specific for vector).

We acknowledge the reviewer’s point that discussion of how “de novo” Cas9-immunity might arise is not included in the paper. Additionally, we acknowledge that the phrase “de novo” may be misleading without clarification so we have removed it from the abstract. However, intraocular infection can activate multiple mechanisms leading to inflammation in the eye, which includes production or concentration of antigen-specific antibodies in the vitreous. Specifically, studies in rabbits have shown that intravitreal injection of cell wall components derived from *S. aureus* induces endophthalmitis (intraocular inflammation) with corresponding

significant increases in vitreous antibody titers against this antigen (PMID: 2016134). Cas9 in the vitreous following successful gene therapy may risk induction of a similar response.

Additionally, we respect the reviewer's important and valid concern regarding overestimating the effect of intraocular antibodies. Given limited mechanistic evidence, we have revised the text to carefully and clearly state that a direct role of antibody-mediated inflammation to artificial Cas9 expression in the retina has not been shown to-date. Our study lays the groundwork for future investigation of this understudied potential risk.

With that being said, while we agree with the reviewer that contributions of humoral immunity to AAV vectors in intraocular inflammation is less well established than cellular immunity, this does not mean that intraocular antibodies are irrelevant. As mentioned above, studies have reported a link between AAV antibody levels and episodes of intraocular inflammation, including vitritis and uveitis (PMID: 25938638, PMID: 30730541, PMID: 28526489, PMID: 28647203). The relevance of humoral immunity to gene therapies is under active investigation.

We have updated the text to address the above concerns.

Critiques pertaining to the Introduction:

6. Critique 6: Line 52/53: authors only refer to own work, not to the results from Simhadri paper, where significantly less a-Cas9 antibodies were reported (5%?).

We have added a comment on this discrepancy to the Introduction, as well as to the Discussion section (see response to Critique 2, above).

7. Critique 7: Line 53/54: cellular immunity was also reported in Wagner et al 2019 (PMID: 30374197) and Ferdosi et al 2019 (PMID: 31015529).

We have updated and expanded the text to include these references.

8. Critique 8: Line 54-65: Wagner et al (PMID: 30374197) showed killing of SpCas9-overexpressing cells by Cas9-specific T cells in vitro, please cite instead of review article (??).

The text has been updated to include this reference.

9. Critique 9: Line 59-61: AAV elicits dose dependent inflammation in ocular gene therapy trials indicating a particular role of innate immune pathways (e.g. cite Chan et al 2021 PMID: 33568518, see Suppl. Fig. 18+19 for results from systemic literature review)

We have included this reference and updated the text to discuss our work more broadly in the context of ongoing ocular gene therapy trials, as well as a discussion of a developing understanding of the relevance of humoral immunity. A full discussion of AAV inflammation lies outside the scope of the present work, which is centered on immune recognition and potential inflammatory risks related to Cas9 protein. It remains unclear whether immune reactions against the genome of a virus thought to be largely non-immunogenic would be the same as immune reactions directed against Cas9, a protein derived from common ocular bacterial pathogens.

10. Critique 10: Line 63/64: please cite original work by Maeder et al (PMID: 30664785), which is the basis of this first clinical trial.

The text has been revised to cite the original work by Maeder et al.

11. Critique 11: line 76/77: true, but is this really mechanistically established?

While no study to-date has shown that Cas9 elicits inflammation in the eye, studies have demonstrated that exposure of retinal and intravitreal antigen to antibodies is linked to pathological inflammation in multiple contexts (PMID: 17235687, PMID: 12848960, PMID: 27428230, PMID: 1566234, PMID: 10865992, PMID: 24334446; see response to critique #1). One established pathological mechanism is formation of antibody-antigen immune complexes, and we have updated the text to state this. Additionally, we have updated the text to avoid overestimating the role of humoral immunity given available evidence (see response to critique #5).

12. Critique 12: Line 79: true + good comment, but toxicity in ocular gene therapy is dose-dependent and relevance of innate immunity is already established (innate immune activation → CD8 T cells lead to elimination of retinal cells).

We acknowledge the reviewer's comment that immune responses to ocular gene therapy (e.g., AAV vectors) has been shown to be mediated by innate immunity in a dose-dependent fashion. We have updated the text to include this perspective and discuss our rationale and results in the context of the field more broadly (see responses to critiques #4 & #5). As stated above, antibody-mediated inflammation has been shown to occur in the eye independent of T-cells, and humoral immunity directed against AAV is linked to intraocular inflammation.

13. Critique 13: Line 89/90: what about rate of anti-AAV antibodies or other anti-vector antibodies? Has there been prior literature that directly links preexisting antibodies to ocular inflammation? —> ref 15 cited in the discussion just looks at the prevalence in corpses.

A discussion of AAV inflammation and mechanisms of antibody-mediated inflammation in the eye has been established and relevant references have been added to the text (see responses to critiques #1, #4, & #5). Antibody responses to viruses in the eye has been well-established, such as for Herpes viruses (PMID: 12852492). As mentioned above, multiple studies have reported a link between increases in systemic AAV antibody levels and development of intraocular inflammation, indicating a relevance for humoral immunity in this context.

Critiques pertaining to the Discussion:

14. Critique 14: would it be possible to use the method described in this paper to evaluate patients undergoing ocular gene therapy? If yes, this could potentially establish the link between anti-Cas9/AAV antibodies and excessive inflammation in the eye after gene therapy.

This is a great comment and yes, we believe this method could be used to evaluate patients receiving gene therapy. Whether or not anti-Cas9 play a significant role in driving inflammation, they may be able to serve as a new and useful biomarker for the presence or extent of gene therapy-driven inflammation. This is further supported by evidence of a close connection between antibody titers and T-cell mediated inflammation even in eye diseases predominantly driven by cellular immunity (PMID: 24315290 PMID: 30635390), as well as by evidence of a link between increases in AAV antibody levels and intraocular inflammation (PMID: 25938638, PMID: 30730541, PMID: 28526489, PMID: 28647203). While these studies have focused on serum antibody measurements, we believe measurements of antibodies in vitreous fluid would be safe and even more relevant given the close association between vitreous fluid and the retina.

15. Critique 15: The authors must discuss why they detect serum a-SpCas9 antibodies at such a high rate, while Simhadri et al detected only 2.5% and Ferdosi et al 8.8% in much larger cohorts? Simhadri reported higher rates for a-SaCas9 ab (10%)

The text has been updated to address this critique. See response to critique #2, above.

16. Critique 16: What is the expected impact of the surgery required for injection of ocular gene therapy? Could transfer of aCas9 antibodies appear later due to lost barrier function?

A loss of blood-retinal-barrier function due to surgical disruption from subretinal injection is important to consider. However, we believe it is unlikely that the mechanical aspects of subretinal injections would

significantly facilitate anti-Cas9 antibody transfer. This is supported through our inclusion of patients with diseases involving damage to the blood retinal barrier (e.g., proliferative diabetic retinopathy). These patients did not show anti-Cas9 positivity in the vitreous despite showing anti-Cas9 positivity in the blood. However, it does remain a possibility (particularly considering the case of a patient with ocular cancer and likely severe blood-retina-barrier disruption, who demonstrated intravitreal anti-Cas9; Supplemental Table 1, Case #8) and the text of the revised Discussion section has been expanded to include this consideration.

Reviewer #2 (Remarks to the Author):

Major concerns:

1. Critique 1: Age of donors is significantly higher compared to the median age of donors tested in Ref 1. In the latter the median age of donors tested was 43. This difference should be taken into consideration in the conclusions.

This is a good comment and we agree with the reviewer that a difference in cohort characteristics is an important consideration that should be addressed in the text. We have expanded our Discussion section to take these cohort differences into consideration. It is unclear to what extent these differences may impact our results. On one hand, the older age of our cohort may be associated with a waning immunity. This may explain the lower raw ELISA values we observed in our cohort compared with Ref 1. However, when these lower values were compared against our control groups, we found a higher overall rate of antibody positivity. On the other hand, the older age of our cohort may also be associated with increased risk or increased lifetime frequency of Staph aureus/Strep pyogenes exposure, potentially leading to a higher prevalence of pre-existing antibodies against these microbes. Although, given the smaller size of our cohort relative to Ref 1, regardless of demographic differences, these differences in prevalence may be due to differences in sample size.

2. Critique 2: The authors need to further substantiate the notion that high enough serum concentrations may become detectable in the eye. This may easily be validated in mice experiencing sepsis due to e.g., S. aureus.

To further address this notion experimentally, we performed a new series of experiments and significantly updated the manuscript. We immunized mice to ovalbumin (positive control), SaCas9, and SpCas9 antigens and used our ELISA platform to detect and quantify antibodies to these antigens in both the serum and vitreous (new Figure 2).

Our data show that we effectively immunized mice to these proteins, generating robust antibody responses in the serum (Figure 2A&B). By 6-weeks post-injection, we also found that a subset of these mice had developed detectable levels of vitreous fluid antibodies against ovalbumin and SpCas9, but less so against SaCas9 (Figure 2C, D, E). This result supports our hypothesis that higher serum concentrations of a particular antibody may lead to its circulation in the eye. Notably, SaCas9 showed the weakest immunization in our mice cohort

(Figure 2B, E), and showed much lower levels of vitreous fluid anti-SaCas9 (10% positive)—unlike what we observed for ovalbumin (40% positive) or SpCas9 (44% positive). We believe this difference is likely due to total circulating concentrations of each of the antibodies, whereas SaCas9 levels in the serum did not meet the threshold to become detectable in the vitreous fluid. Furthermore, we again observed a trend across all immunization conditions where mice with higher levels of a particular serum antibody tended to show higher levels of that antibody in the vitreous as well (Figure 2F).

Further studies are needed to address this hypothesis more thoroughly, but our new mouse dataset provides additional evidence supporting the notion that serum antibody concentration is likely a key determinant in whether those same antibodies may become detectable in the vitreous fluid.

The revised manuscript has been updated to include this new experiment (Figure 2, Supplemental Dataset 3) designed to address this reviewer comment.

3. Critique 3: It would strengthen the paper to include analysis of immune reactions (paired vitreous-serum biopsies) in e.g., mice (with or without preexisting Cas9 antibodies) following viral vector-based ocular delivery of clinically relevant levels of Cas9. The delivery route (subretinal or intravitreal) may also have impact on the immune response.

This is an excellent suggestion by the reviewer. As mentioned in response to Critique 2, above, we have begun to address this notion experimentally in mice and noted this point in the discussion. While we did not use viral-vector-based ocular delivery of clinically relevant levels of Cas9, this is an active area of future investigation for our lab. We believe that the fully described experiment lies outside the scope of the present paper.

4. Critique 4: As stated, up to 58-78 % of the general population may exhibit preexisting Cas9 ab. Even though the focus of the present study is the eye, please include a brief discussion on the high abundance of preexisting Cas9 antibodies in relation to the relevance of Cas9-based gene therapy in tissues that are not immune privileged.

We appreciate the reviewer's perspective and agree that a discussion of the high prevalence of anti-Cas9 in the serum systemically would be an important addition to the text.

Unlike in the eye, systemic tissues are subject to regular surveillance by a wide variety of antibodies and T-cells. In this regard, pre-existing T-cell populations may represent a greater risk to systemic tissues than antibodies (PMID: 30158648). However, a risk still exists for the formation of immune complexes consisting of Cas9 antigen bound to preexisting circulating antibodies, a well-described phenomenon in a variety of human diseases (PMID: 6157327). This risk would likely depend on the amount of Cas9 being expressed by edited cells, the duration of expression needed to achieve therapeutic efficacy, and the potential for extracellular Cas9

leak from edited tissue. Separately, the presence of Cas9 antibodies systemically likely also serves as a useful biomarker for the risk of an immune response overall. It is likely that levels of Cas9 antibodies correlate with this risk in a way that could be used to evaluate potential gene therapy candidates prior to and during therapy.

We have added additional text on this perspective to the revised Discussion section.

Minor concerns:

5. Critique 5: The title is misleading since the main finding is that Cas9 antibodies apparently are not present in the eye. Please rephrase and include “human”. Alternatively, include “Human” and “?” in the submitted version: Cas9 Antibodies in the Human Eye?

We have updated the title to, “Investigation of Cas9 Antibodies in the Human Eye.”

6. Critique 6: Is Reference NCT02168686 correct? This is a gene therapy trial to treat A1AT deficiency. Maybe I missed it, but it is not obvious from the web page how the authors conclude from this reference that immune reactions to gene therapy in the eye have halted trials. Please verify.

We have included the correct references (PMID: 25938638, PMID: 29940166, PMID: 30297895) and updated the revised text accordingly.

7. Critique 7: Page 5, Results. Please indicate which cohort the sample originate from. Moreover, the “number of vitreous biopsies” is used inconsistently: In the Figure 1 legend it listed as 26, in the Supplementary Methods, it is 28. Please verify.

We have updated the text and legends to consistently state the correct cohort size (n=28).

8. Critique 8: Line 99, “approximately 150-fold fewer”. From Figure 1B up to 200-fold fewer total antibodies can be observed. Range should be indicated.

This is a good observation by the reviewer. For clarity, we have updated the text to state that, overall, we detected vitreous fluid antibodies at a vitreous to serum ratio of approximately 1:143, with sample-to-sample differences. We observed no obvious correlations with patient diagnoses. For each individual antibody subclass, we detected the following approximate ratios of vitreous fluid to serum antibody concentrations, with vitreous antibody ranges and standard deviations reported in parentheses: IgG₁ = 1:150 (range: 19.05 mg/dL,

s.d. 4.93), IgG₂ = 1:133 (range: 16.86 mg/dL, s.d. 4.43), IgG₃ = 1:205 (range: 2.69 mg/dL, s.d. 0.71), IgG₄ = 1:13 (range: 12.72 mg/dL, s.d. 4.19), IgM = 1:926 (range: 2.40 mg/dL, s.d. 0.45), IgA = 1:230 (range: 8.61 mg/dL, s.d. 1.98).

9. Critique 9: For clarity use vitreous fluid in the manuscript. Not intraocular fluid.

We have updated the text to more accurately state 'vitreous fluid' throughout the manuscript.

10. Critique 10: Regarding Suppl Table 1: Please stratify the cohort information, e.g., add some descriptive features as gender ratio and median age.

We have stratified the cohort information in Supplemental Table 1 and added descriptive features, including the gender ratio and median age.

11. Critique 11: Will the modified CRISPR/Cas tools, which have emerged or are on its way, attract attention from the immune system similar to SaCas9 and SpCas9?

This is a great comment by the reviewer and we have added text on modified CRISPR/Cas9 tools to the Discussion.

If immune reaction to SaCas9/SpCas9 proves to be an ongoing hurdle, there are multiple options to attempt to circumvent this. One option is to use Cas9 protein derived from microbes that do not colonize humans, such as the thermophilic bacterium *Bacillus hisashi* (PMID: 30670702). In this case, the use of *B. hisashi* Cas12b may represent less of an inflammatory risk due to absence of pre-existing immunity. Another similar alternative Cas9 might be derived from *Geobacillus stearthermophilus* (PMID: 29127284). However, it is still possible that these proteins might be recognized by the human body as foreign and remain capable of generating an immune response. Alternatively, another option may be to engineer forms of Cas9 that are designed to be less-immunogenic through silencing of immunodominant epitopes (PMID: 31015529), or through inclusion of oligonucleotides which directly antagonize relevant immune receptors (PMID: 33568518). Separately, a third route to circumvent this issue might involve inducing immune tolerance, perhaps through methods of expanding the Cas9-specific regulatory T-cell population (PMID: 30374197, PMID: 31589876).

We have updated the text to include this discussion.

Reviewers' Comments:

Reviewer #1:

Remarks to the Author:

The authors satisfactorily addressed all of my concerns.

Reviewer #2:

Remarks to the Author:

All my concerns have been addressed in the revision.

Response to Reviewer #1 (Remarks to the Author):

General critiques:

1. Critique 1: Conceptionally, the role of a-Cas9 antibodies may aggravate inflammatory responses and prior literature in ocular gene therapy has shown dose-dependent inflammation triggered by AAV alone - most probably due to innate immune responses to the vector genome which attract CD8+ T cells. Therefore, the detection of a-Cas9 antibodies in the eye only poses a theoretical risk, while preexisting Cas9-specific T cells would be more detrimental.

We thank the reviewer for this insightful comment and agree that detection of preexisting Cas9-specific T-cells in the eye would represent a significant risk to edited retinal cells, and that T-cells may be primary effector cells in some retinal immunopathologies including AAV-triggered inflammation. However, T-cells are normally absent from the eye, in particular from the vitreous fluid and subretinal space, and their specific role in various retinal inflammation conditions is controversial. On the other hand, antibodies are normally circulating in the vitreous fluid and sample the retina space, and preexisting antibodies to antigens are a well-established concern. The contribution and association of antibodies to retinal pathology has been described in multiple contexts, including AAV-related inflammation. While exact mechanisms of antibody function in the eye are unclear, the potential for formation of immune complexes consisting of Cas9 antigen bound to preexisting Cas9 antibodies represents at least one inflammatory mechanism which threatens the delicate intraocular environment independent of preexisting T-cells.

With regards to harmful consequences of antibodies in the eye, studies of autoimmune retinopathy have shown that exposing retinal cells to autoantibodies *in vitro* can induce their apoptosis, and a similar phenomenon has been observed *in vivo* following intravitreal injection of autoantibodies into rat eyes (PMID: 17235687, PMID: 12848960). While anti-Cas9 are not necessarily akin to autoantibodies, this finding demonstrates the immunopathological risk of intraocular antibodies independent of T-cells. Furthermore, formation of deleterious extracellular antibody-antigen immune complexes has been demonstrated in the context of eye disease. This mechanism is more closely relevant to potential inflammatory consequences of anti-Cas9, whereas anti-Cas9 might form immune complexes with extracellular Cas9 in the vitreous, shed from edited cells. Indeed, the presence of antibody-antigen immune complexes is strongly associated with development of uveitis in humans (PMID: 27428230), and immune complexes are thought to play a significant and direct role in the pathology of lens-induced endophthalmitis, for example (PMID: 1566234). Furthermore, in the context of age-related macular degeneration, immune complex formation may play a significant role in the development of drusen (PMID: 10865992). Thus, recognition of ocular Cas9 antigen by preexisting antibodies risks formation of harmful immune complexes. These immune complexes have the potential to bind Fcγ receptors expressed on the surface of microglia and other retinal cells, driving inflammation. Accordingly, intravitreal injection of antigen in immunized mice has been shown to generate antibody-antigen immune complexes throughout the retina, including large deposits in the subretinal space, leading to a potent inflammatory response involving activation of microglia, macrophages, and expression of pro-inflammatory cytokines (PMID: 24334446). These antibody-dependent inflammatory mechanisms do not involve T-cells.

While T-cells are thought to be drivers of intraocular inflammation, they are normally absent from the eye—particularly the posterior compartment where the retina is located. On the other hand, antibodies, cytokines, complement molecules, and resident phagocytic cells have a permanent presence in the eye. Thus, it is thought that the main function of intraocular antibodies is to clear infections and limit the spread of pathogens while innate and adaptive cell-mediated immunity remains suppressed (PMID: 12852492). Yet, while antibodies appear capable of inducing intraocular inflammation independent of T-cells, antibody-mediated inflammation is likely also synergistic with T-cells, facilitating T-cell infiltration and potentiating their activities. Studies in mouse models of autoimmune retinopathy have shown that even when the blood-retina-barrier is experimentally broken to facilitate T-cell entry, removal of antibodies results in slower disease progression and a milder form of the disease (PMID: 30635390). Furthermore, while AAV-mediated inflammation can be driven by T-cells, studies have reported a link between systemic AAV antibody levels and episodes of intraocular inflammation, including vitritis and uveitis (PMID: 25938638, PMID: 30730541, PMID: 28526489, PMID: 28647203). Accordingly, even if anti-Cas9 itself does not directly drive disease, its presence may still serve as a useful and novel biomarker for predicting the potential for immune response. This is also supported by the strong correlation between the presence of anti-retinal antibodies and the development of autoimmune retinopathy, even though the disease is mediated by T-cells (PMID: 24315290)

Ocular immune responses are complex and remain incompletely understood, but they are likely mediated by multiple inflammatory mechanisms. Evidence indicates that T-cells are key drivers of ocular inflammation, but the role of antibodies and immune complexes in the eye is particularly poorly understood. The ubiquity of antibodies within ocular tissues and evidence of a direct pathological role in some contexts indicates that they are an important component to consider and likely also serve as a sensitive biomarker for gauging overall immunosurveillance.

Based on the reviewer's important comment, we have updated the text to better reflect this perspective, the limitations of our study, and recognition of the potential role of T-cells.

2. Critique 2: The discussion seems circular and repetitive and there is little discussion of prior data on Cas9-immunity by other groups (only references to own work), in particular no comment on discrepancy between a-Cas9 antibody prevalence between original Cas9-immunity paper by this group and prior work by Simhardi et al (PMID: 30073181).

We agree with the reviewer's astute comment that this discrepancy is critical to discuss, and we thank them for this suggestion. As such, we have updated the text of the Discussion to comment on these discrepancies in the context of the field and have included references to other important studies in the field, including the work by Simhardi *et al.* and have updated the discussion to remove repetitive text, as described below.

The reviewer raises an important point regarding discrepancies between studies investigating the prevalence of anti-Cas9 in human donors. While data from our detection platform show much higher anti-Cas9 prevalence than the work by Simhardi *et al.*, and we have added this reference, we believe these differences may be due to higher sensitivity of our platform. Furthermore, our results show high correlation with clinical data and remain closely aligned with reported frequencies of cellular immunity in donor samples, supporting our conclusions.

Using our platform, previous work detected antibodies against *S. pyogenes* Cas9 (SpCas9) and *S. aureus* (SaCas9) in 58% and 78% of human donor serum samples, respectively (PMID: 30692695). In the present study, we again identified high prevalence of these antibodies, with 100% of our 13 paired serum samples positive for both anti-SpCas9 and anti-SaCas9. These results differ from those in the mentioned study by Simhardi *et al.* (PMID: 30073181) as well as a study by Ferdosi *et al.* (PMID: 31015529), where only 2.5% (10% for SpCas9) and at least 5% of donor serum samples were called positive for anti-Cas9, respectively. However, both Ferdosi *et al.* and Wagner *et al.* detected T-cells reactive to Cas9 at frequencies of 60% or higher, indicating that the human immune system is exposed to and develops a response to Cas9 regularly (PMID: 31015529, PMID: 30374197). With regards to circulating anti-Cas9 levels, we believe that large differences in prevalence reported between studies are due to relatively low levels of circulating anti-Cas9 (even in positive individuals) and differences in sensitivity between detection platforms. We added this explanation to the discussion text.

In this regard, we believe that our platform is more sensitive—a conclusion supported by evidence of our positive control anti-tetanus signal maxing out the detector in our serum samples (Fig 1C). While this high sensitivity may indicate the potential for false-positives, the validity of antibody frequencies reported in our study is supported by the tractability of our system (e.g., including a positive control vitreous sample from a patient with intraocular bacterial infection and seeing a corresponding increase in anti-Cas9 levels, as shown in Fig 1F), as well as patient-to-patient and clinical correlations in our data (e.g., higher serum anti-Cas9 tending to predict higher vitreous anti-Cas9 [Fig 1E]). With regards to the latter point, in Fig 1D, both positive anti-Cas9 vitreous samples came from the same patient with ocular cancer (Supplemental Table 1, Case #8)—a condition showing major anatomical disruption to the eye and severe damage to the blood-vitreous barrier. As expected, this patient showed higher anti-Cas9 antibody levels, likely representing significant leak from the serum into the vitreous, a finding which was observed due to the sensitivity of our platform.

Furthermore, our reported frequencies of anti-Cas9 in human donor serum are more closely aligned with reported frequencies of Cas9-reactive T-cells (cellular immunity) in human donor serum. This consistency with reported frequencies of cellular immunity strengthens the validity of our platform. Specifically, Ferdosi *et al.* found Cas9-reactive T-cells in the majority of their healthy cohort (PMID: 31015529) and Wagner *et al.* found 96% of their donor samples to be positive for Cas9-reactive T-cells (PMID: 30374197). Thus, while mechanisms of cellular and humoral immunity differ, it is not unrealistic to conclude that antibody frequencies may occur at similar levels. Of course, further research by independent groups is needed to fully resolve these discrepancies in antibody frequencies.

Finally (and importantly), despite the likely high sensitivity of our detection platform, we were unable to detect high frequencies of anti-Cas9 in the vitreous of the eye despite detecting high frequencies of control anti-Tetanus antibodies in the eye. This supports the core overall conclusion of the present work that preexisting immunity to Cas9 in the serum does not occur at the same frequency in the eye.

3. Critique 3: I rather recommend the work for a brief report (more suitable as “brief communication” instead of a full article).

Per both reviewer’s suggestions, we have significantly expanded the text, as well as added an additional mouse dataset and figure along with additional supplemental data to the manuscript (Figure 2, Supplemental Data 3). We will work with the journal editors to determine the best format.

Critiques pertaining to the Abstract:

4. Critique 4: Line 39/40: authors infer a logical connection between antibodies and ocular inflammation after gene therapy, which is not established as a mechanism in the field to my knowledge.

This is a fair point, as mechanisms of antibodies in the eye are poorly understood, and we have adjusted the words to acknowledge this. However, multiple studies have reported a correlation between levels of neutralizing antibody levels to AAV gene therapies and development of intraocular inflammation (PMID: 25938638, PMID: 30730541, PMID: 28526489, PMID: 28647203). Furthermore, we believe there is a logical connection between antibodies and ocular inflammation to Cas9 protein, a bacterial protein that does correlate with a case of intraocular infection, rather than inflammation specific to gene therapy more broadly. The Cas9 protein is derived from a known immunogenic ocular pathogen. Given the role of antibodies in other ocular pathologies (see response to Critique #1, above), we believe that our study adds value and a new perspective to the field through interrogation of these antibodies.

5. Critique 5: Line 45/46: how does de novo intraocular Cas9-immunity arise? Not discussed in the paper: immune privilege = little amounts of adaptive immune cells around, please delete this comment, because it is overestimating the role of humoral anti-Cas9 immunity and is not in line with current understanding how AAV elicits inflammation (which is the activation of innate immune response by vector genomes, which then attracts T cells specific for vector).

We acknowledge the reviewer’s point that discussion of how “de novo” Cas9-immunity might arise is not included in the paper. Additionally, we acknowledge that the phrase “de novo” may be misleading without clarification so we have removed it from the abstract. However, intraocular infection can activate multiple mechanisms leading to inflammation in the eye, which includes production or concentration of antigen-specific antibodies in the vitreous. Specifically, studies in rabbits have shown that intravitreal injection of cell wall components derived from *S. aureus* induces endophthalmitis (intraocular inflammation) with corresponding

significant increases in vitreous antibody titers against this antigen (PMID: 2016134). Cas9 in the vitreous following successful gene therapy may risk induction of a similar response.

Additionally, we respect the reviewer's important and valid concern regarding overestimating the effect of intraocular antibodies. Given limited mechanistic evidence, we have revised the text to carefully and clearly state that a direct role of antibody-mediated inflammation to artificial Cas9 expression in the retina has not been shown to-date. Our study lays the groundwork for future investigation of this understudied potential risk.

With that being said, while we agree with the reviewer that contributions of humoral immunity to AAV vectors in intraocular inflammation is less well established than cellular immunity, this does not mean that intraocular antibodies are irrelevant. As mentioned above, studies have reported a link between AAV antibody levels and episodes of intraocular inflammation, including vitritis and uveitis (PMID: 25938638, PMID: 30730541, PMID: 28526489, PMID: 28647203). The relevance of humoral immunity to gene therapies is under active investigation.

We have updated the text to address the above concerns.

Critiques pertaining to the Introduction:

6. Critique 6: Line 52/53: authors only refer to own work, not to the results from Simhadri paper, where significantly less a-Cas9 antibodies were reported (5%?).

We have added a comment on this discrepancy to the Introduction, as well as to the Discussion section (see response to Critique 2, above).

7. Critique 7: Line 53/54: cellular immunity was also reported in Wagner et al 2019 (PMID: 30374197) and Ferdosi et al 2019 (PMID: 31015529).

We have updated and expanded the text to include these references.

8. Critique 8: Line 54-65: Wagner et al (PMID: 30374197) showed killing of SpCas9-overexpressing cells by Cas9-specific T cells in vitro, please cite instead of review article (??).

The text has been updated to include this reference.

9. Critique 9: Line 59-61: AAV elicits dose dependent inflammation in ocular gene therapy trials indicating a particular role of innate immune pathways (e.g. cite Chan et al 2021 PMID: 33568518, see Suppl. Fig. 18+19 for results from systemic literature review)

We have included this reference and updated the text to discuss our work more broadly in the context of ongoing ocular gene therapy trials, as well as a discussion of a developing understanding of the relevance of humoral immunity. A full discussion of AAV inflammation lies outside the scope of the present work, which is centered on immune recognition and potential inflammatory risks related to Cas9 protein. It remains unclear whether immune reactions against the genome of a virus thought to be largely non-immunogenic would be the same as immune reactions directed against Cas9, a protein derived from common ocular bacterial pathogens.

10. Critique 10: Line 63/64: please cite original work by Maeder et al (PMID: 30664785), which is the basis of this first clinical trial.

The text has been revised to cite the original work by Maeder et al.

11. Critique 11: line 76/77: true, but is this really mechanistically established?

While no study to-date has shown that Cas9 elicits inflammation in the eye, studies have demonstrated that exposure of retinal and intravitreal antigen to antibodies is linked to pathological inflammation in multiple contexts (PMID: 17235687, PMID: 12848960, PMID: 27428230, PMID: 1566234, PMID: 10865992, PMID: 24334446; see response to critique #1). One established pathological mechanism is formation of antibody-antigen immune complexes, and we have updated the text to state this. Additionally, we have updated the text to avoid overestimating the role of humoral immunity given available evidence (see response to critique #5).

12. Critique 12: Line 79: true + good comment, but toxicity in ocular gene therapy is dose-dependent and relevance of innate immunity is already established (innate immune activation → CD8 T cells lead to elimination of retinal cells).

We acknowledge the reviewer's comment that immune responses to ocular gene therapy (e.g., AAV vectors) has been shown to be mediated by innate immunity in a dose-dependent fashion. We have updated the text to include this perspective and discuss our rationale and results in the context of the field more broadly (see responses to critiques #4 & #5). As stated above, antibody-mediated inflammation has been shown to occur in the eye independent of T-cells, and humoral immunity directed against AAV is linked to intraocular inflammation.

13. Critique 13: Line 89/90: what about rate of anti-AAV antibodies or other anti-vector antibodies? Has there been prior literature that directly links preexisting antibodies to ocular inflammation? —> ref 15 cited in the discussion just looks at the prevalence in corpses.

A discussion of AAV inflammation and mechanisms of antibody-mediated inflammation in the eye has been established and relevant references have been added to the text (see responses to critiques #1, #4, & #5). Antibody responses to viruses in the eye has been well-established, such as for Herpes viruses (PMID: 12852492). As mentioned above, multiple studies have reported a link between increases in systemic AAV antibody levels and development of intraocular inflammation, indicating a relevance for humoral immunity in this context.

Critiques pertaining to the Discussion:

14. Critique 14: would it be possible to use the method described in this paper to evaluate patients undergoing ocular gene therapy? If yes, this could potentially establish the link between anti-Cas9/AAV antibodies and excessive inflammation in the eye after gene therapy.

This is a great comment and yes, we believe this method could be used to evaluate patients receiving gene therapy. Whether or not anti-Cas9 play a significant role in driving inflammation, they may be able to serve as a new and useful biomarker for the presence or extent of gene therapy-driven inflammation. This is further supported by evidence of a close connection between antibody titers and T-cell mediated inflammation even in eye diseases predominantly driven by cellular immunity (PMID: 24315290 PMID: 30635390), as well as by evidence of a link between increases in AAV antibody levels and intraocular inflammation (PMID: 25938638, PMID: 30730541, PMID: 28526489, PMID: 28647203). While these studies have focused on serum antibody measurements, we believe measurements of antibodies in vitreous fluid would be safe and even more relevant given the close association between vitreous fluid and the retina.

15. Critique 15: The authors must discuss why they detect serum a-SpCas9 antibodies at such a high rate, while Simhadri et al detected only 2.5% and Ferdosi et al 8.8% in much larger cohorts? Simhadri reported higher rates for a-SaCas9 ab (10%)

The text has been updated to address this critique. See response to critique #2, above.

16. Critique 16: What is the expected impact of the surgery required for injection of ocular gene therapy? Could transfer of aCas9 antibodies appear later due to lost barrier function?

A loss of blood-retinal-barrier function due to surgical disruption from subretinal injection is important to consider. However, we believe it is unlikely that the mechanical aspects of subretinal injections would

significantly facilitate anti-Cas9 antibody transfer. This is supported through our inclusion of patients with diseases involving damage to the blood retinal barrier (e.g., proliferative diabetic retinopathy). These patients did not show anti-Cas9 positivity in the vitreous despite showing anti-Cas9 positivity in the blood. However, it does remain a possibility (particularly considering the case of a patient with ocular cancer and likely severe blood-retina-barrier disruption, who demonstrated intravitreal anti-Cas9; Supplemental Table 1, Case #8) and the text of the revised Discussion section has been expanded to include this consideration.

Reviewer #2 (Remarks to the Author):

Major concerns:

1. Critique 1: Age of donors is significantly higher compared to the median age of donors tested in Ref 1. In the latter the median age of donors tested was 43. This difference should be taken into consideration in the conclusions.

This is a good comment and we agree with the reviewer that a difference in cohort characteristics is an important consideration that should be addressed in the text. We have expanded our Discussion section to take these cohort differences into consideration. It is unclear to what extent these differences may impact our results. On one hand, the older age of our cohort may be associated with a waning immunity. This may explain the lower raw ELISA values we observed in our cohort compared with Ref 1. However, when these lower values were compared against our control groups, we found a higher overall rate of antibody positivity. On the other hand, the older age of our cohort may also be associated with increased risk or increased lifetime frequency of Staph aureus/Strep pyogenes exposure, potentially leading to a higher prevalence of pre-existing antibodies against these microbes. Although, given the smaller size of our cohort relative to Ref 1, regardless of demographic differences, these differences in prevalence may be due to differences in sample size.

2. Critique 2: The authors need to further substantiate the notion that high enough serum concentrations may become detectable in the eye. This may easily be validated in mice experiencing sepsis due to e.g., S. aureus.

To further address this notion experimentally, we performed a new series of experiments and significantly updated the manuscript. We immunized mice to ovalbumin (positive control), SaCas9, and SpCas9 antigens and used our ELISA platform to detect and quantify antibodies to these antigens in both the serum and vitreous (new Figure 2).

Our data show that we effectively immunized mice to these proteins, generating robust antibody responses in the serum (Figure 2A&B). By 6-weeks post-injection, we also found that a subset of these mice had developed detectable levels of vitreous fluid antibodies against ovalbumin and SpCas9, but less so against SaCas9 (Figure 2C, D, E). This result supports our hypothesis that higher serum concentrations of a particular antibody may lead to its circulation in the eye. Notably, SaCas9 showed the weakest immunization in our mice cohort

(Figure 2B, E), and showed much lower levels of vitreous fluid anti-SaCas9 (10% positive)—unlike what we observed for ovalbumin (40% positive) or SpCas9 (44% positive). We believe this difference is likely due to total circulating concentrations of each of the antibodies, whereas SaCas9 levels in the serum did not meet the threshold to become detectable in the vitreous fluid. Furthermore, we again observed a trend across all immunization conditions where mice with higher levels of a particular serum antibody tended to show higher levels of that antibody in the vitreous as well (Figure 2F).

Further studies are needed to address this hypothesis more thoroughly, but our new mouse dataset provides additional evidence supporting the notion that serum antibody concentration is likely a key determinant in whether those same antibodies may become detectable in the vitreous fluid.

The revised manuscript has been updated to include this new experiment (Figure 2, Supplemental Dataset 3) designed to address this reviewer comment.

3. Critique 3: It would strengthen the paper to include analysis of immune reactions (paired vitreous-serum biopsies) in e.g., mice (with or without preexisting Cas9 antibodies) following viral vector-based ocular delivery of clinically relevant levels of Cas9. The delivery route (subretinal or intravitreal) may also have impact on the immune response.

This is an excellent suggestion by the reviewer. As mentioned in response to Critique 2, above, we have begun to address this notion experimentally in mice and noted this point in the discussion. While we did not use viral-vector-based ocular delivery of clinically relevant levels of Cas9, this is an active area of future investigation for our lab. We believe that the fully described experiment lies outside the scope of the present paper.

4. Critique 4: As stated, up to 58-78 % of the general population may exhibit preexisting Cas9 ab. Even though the focus of the present study is the eye, please include a brief discussion on the high abundance of preexisting Cas9 antibodies in relation to the relevance of Cas9-based gene therapy in tissues that are not immune privileged.

We appreciate the reviewer's perspective and agree that a discussion of the high prevalence of anti-Cas9 in the serum systemically would be an important addition to the text.

Unlike in the eye, systemic tissues are subject to regular surveillance by a wide variety of antibodies and T-cells. In this regard, pre-existing T-cell populations may represent a greater risk to systemic tissues than antibodies (PMID: 30158648). However, a risk still exists for the formation of immune complexes consisting of Cas9 antigen bound to preexisting circulating antibodies, a well-described phenomenon in a variety of human diseases (PMID: 6157327). This risk would likely depend on the amount of Cas9 being expressed by edited cells, the duration of expression needed to achieve therapeutic efficacy, and the potential for extracellular Cas9

leak from edited tissue. Separately, the presence of Cas9 antibodies systemically likely also serves as a useful biomarker for the risk of an immune response overall. It is likely that levels of Cas9 antibodies correlate with this risk in a way that could be used to evaluate potential gene therapy candidates prior to and during therapy.

We have added additional text on this perspective to the revised Discussion section.

Minor concerns:

5. Critique 5: The title is misleading since the main finding is that Cas9 antibodies apparently are not present in the eye. Please rephrase and include “human”. Alternatively, include “Human” and “?” in the submitted version: Cas9 Antibodies in the Human Eye?

We have updated the title to, “Investigation of Cas9 Antibodies in the Human Eye.”

6. Critique 6: Is Reference NCT02168686 correct? This is a gene therapy trial to treat A1AT deficiency. Maybe I missed it, but it is not obvious from the web page how the authors conclude from this reference that immune reactions to gene therapy in the eye have halted trials. Please verify.

We have included the correct references (PMID: 25938638, PMID: 29940166, PMID: 30297895) and updated the revised text accordingly.

7. Critique 7: Page 5, Results. Please indicate which cohort the sample originate from. Moreover, the “number of vitreous biopsies” is used inconsistently: In the Figure 1 legend it listed as 26, in the Supplementary Methods, it is 28. Please verify.

We have updated the text and legends to consistently state the correct cohort size (n=26).

8. Critique 8: Line 99, “approximately 150-fold fewer”. From Figure 1B up to 200-fold fewer total antibodies can be observed. Range should be indicated.

This is a good observation by the reviewer. For clarity, we have updated the text to state that, overall, we detected vitreous fluid antibodies at a vitreous to serum ratio of approximately 1:143, with sample-to-sample differences. We observed no obvious correlations with patient diagnoses. For each individual antibody subclass, we detected the following approximate ratios of vitreous fluid to serum antibody concentrations, with vitreous antibody ranges and standard deviations reported in parentheses: IgG₁ = 1:150 (range: 19.05 mg/dL,

s.d. 4.93), IgG₂ = 1:133 (range: 16.86 mg/dL, s.d. 4.43), IgG₃ = 1:205 (range: 2.69 mg/dL, s.d. 0.71), IgG₄ = 1:13 (range: 12.72 mg/dL, s.d. 4.19), IgM = 1:926 (range: 2.40 mg/dL, s.d. 0.45), IgA = 1:230 (range: 8.61 mg/dL, s.d. 1.98).

9. Critique 9: For clarity use vitreous fluid in the manuscript. Not intraocular fluid.

We have updated the text to more accurately state 'vitreous fluid' throughout the manuscript.

10. Critique 10: Regarding Suppl Table 1: Please stratify the cohort information, e.g., add some descriptive features as gender ratio and median age.

We have stratified the cohort information in Supplemental Table 1 and added descriptive features, including the gender ratio and median age.

11. Critique 11: Will the modified CRISPR/Cas tools, which have emerged or are on its way, attract attention from the immune system similar to SaCas9 and SpCas9?

This is a great comment by the reviewer and we have added text on modified CRISPR/Cas9 tools to the Discussion.

If immune reaction to SaCas9/SpCas9 proves to be an ongoing hurdle, there are multiple options to attempt to circumvent this. One option is to use Cas9 protein derived from microbes that do not colonize humans, such as the thermophilic bacterium *Bacillus hisashi* (PMID: 30670702). In this case, the use of *B. hisashi* Cas12b may represent less of an inflammatory risk due to absence of pre-existing immunity. Another similar alternative Cas9 might be derived from *Geobacillus stearthermophilus* (PMID: 29127284). However, it is still possible that these proteins might be recognized by the human body as foreign and remain capable of generating an immune response. Alternatively, another option may be to engineer forms of Cas9 that are designed to be less-immunogenic through silencing of immunodominant epitopes (PMID: 31015529), or through inclusion of oligonucleotides which directly antagonize relevant immune receptors (PMID: 33568518). Separately, a third route to circumvent this issue might involve inducing immune tolerance, perhaps through methods of expanding the Cas9-specific regulatory T-cell population (PMID: 30374197, PMID: 31589876).

We have updated the text to include this discussion.